# Single-cell RNA sequencing reveals time- and sex-specific responses of mouse spinal cord microglia to peripheral nerve injury and links ApoE to chronic pain

Shannon Tansley[1,2,16], Sonali Uttam[1,16], Alba Ureña Guzmán [1,16], Moein Yaqubi [3], Alain Pacis[4], Marc Parisien [1,5], Haley Deamond [6], Calvin Wong [1], Oded Rabau[7], Nicole Brown[1], Lisbet Haglund [8], Jean Ouellet[7], Carlo Santaguida[9], Alfredo Ribeiro-da-Silva [5,6], Soroush Tahmasebi[10], Masha Prager-Khoutorsky [11], Jiannis Ragoussis [12,13], Ji Zhang[5,14], Michael W. Salter [15], Luda Diatchenko [1,5], Luke M. Healy[3✉], Jeffrey S. Mogil [1,2,5✉] & Arkady Khoutorsky [1,5✉]

Activation of microglia in the spinal cord following peripheral nerve injury is critical for the development of long-lasting pain hypersensitivity. However, it remains unclear whether distinct microglia subpopulations or states contribute to different stages of pain development and maintenance. Using single-cell RNA-sequencing, we show that peripheral nerve injury induces the generation of a male-specific inflammatory microglia subtype, and demonstrate increased proliferation of microglia in male as compared to female mice. We also show time- and sex-specific transcriptional changes in different microglial subpopulations following peripheral nerve injury. Apolipoprotein E (*Apoe*) is the top upregulated gene in spinal cord microglia at chronic time points after peripheral nerve injury in mice. Furthermore, polymorphisms in the *APOE* gene in humans are associated with chronic pain. Single-cell RNA sequencing analysis of human spinal cord microglia reveals a subpopulation with a disease-related transcriptional signature. Our data provide a detailed analysis of transcriptional states of mouse and human spinal cord microglia, and identify a link between ApoE and chronic pain in humans.

[1] Department of Anesthesia and Faculty of Dental Medicine and Oral Health Sciences, McGill University, Montréal, QC, Canada. [2] Department of Psychology, McGill University, Montréal, QC, Canada. [3] Neuroimmunology Unit, Montreal Neurological Institute, McGill University, Québec, Canada. [4] Canadian Centre for Computational Genomics, McGill Genome Centre, Montréal, QC, Canada. [5] Alan Edwards Centre for Research on Pain, McGill University, Montréal, QC, Canada. [6] Department of Pharmacology and Therapeutics, McGill University, Montreal, Canada. [7] McGill Scoliosis and Spine Group, Department of Surgery, McGill University, Montreal, QC, Canada. [8] The Orthopaedic Research Laboratory, Department of Surgery, McGill University, Montreal, Canada. [9] Department of Neurology & Neurosurgery, Montreal Neurological Institute, McGill University, Montreal, Canada. [10] Department of Pharmacology and Regenerative Medicine, University of Illinois at Chicago, Chicago, IL 60612, USA. [11] Department of Physiology, McGill University, Montreal, QC, Canada. [12] Department of Human Genetics and Department of Bioengineering, McGill University, Montreal, QC, Canada. [13] McGill University Genome Centre, Montreal, QC, Canada. [14] Department of Neurology & Neurosurgery and Faculty of Dentistry, McGill University, Montréal, QC, Canada. [15] Neurosciences & Mental Health Program, Hospital for Sick Children, Department of Physiology, Faculty of Medicine, University of Toronto, Toronto, ON, Canada. [16]These authors contributed equally: Shannon Tansley, Sonali Uttam, Alba Ureña Guzmán. ✉email: luke.healy@mcgill.ca; jeffrey.mogil@mcgill.ca; arkady.khoutorsky@mcgill.ca

Peripheral nerve injury often leads to neuropathic pain, a debilitating condition associated with spontaneous and light touch-induced pain, and accompanied by the activation and proliferation of microglia in the spinal cord dorsal horn[1,2]. Microglia are resident immune cells in the central nervous system (CNS) that constantly survey the environment. Damaged primary afferents release multiple molecules such as chemokines (CCL2, CCL21, CXCL1), signaling molecules (CSF1), and proteases (MMP-9) to induce microglia proliferation and activation, which is characterised by a transition from a ramified homeostatic microglia phenotype with multiple fine processes to an amoeboid-like cell[3]. The "activated" microglia release a wide range of bioactive substances that signal to neuronal and non-neuronal cells to facilitate pain transmission, critically contributing to pain hypersensitivity[4–7]. Microglia play important roles in both the development and maintenance phases of neuropathic pain as inhibition or depletion of microglia 1–2 weeks or 3 months post-peripheral nerve injury alleviate hypersensitivity[5,8,9]. The role of microglia in neuropathic pain is known to be sex-dependent[10,11]. Microglia proliferation and morphological changes in response to peripheral nerve injury are present in both sexes; however, a functional role of microglia as critical drivers of neuropathic pain is observed in male but not in female animals[12–16].

Progress in single-cell transcriptomic techniques has facilitated the study of microglia heterogeneity in different physiological and pathological processes, revealing that microglia in the brain exist in multiple transcriptional states tailored to different developmental stages, anatomical areas, and pathologies[17–20]. Nine distinct transcriptional states of microglia have been described in the brain[17], including several subpopulations of homeostatic microglia, and microglia responding to injury (injury-responsive microglia; IRM), and to chronic pathologies including Alzheimer's disease (AD) (disease-associated microglia; DAM)[21]. The heterogeneity of microglia in the mouse and human spinal cord, however, remains ill-defined. Despite the central roles of microglia in the development and maintenance of neuropathic pain[3], it is unknown whether peripheral nerve injury leads to the generation of unique microglia transcriptional states that promote and maintain pain hypersensitivity.

In this study, we used single-cell RNA sequencing (scRNA-seq) to define transcriptional states of spinal cord microglia in mouse and human. In mice, by profiling a large number (188,787) of cells in both sexes, we observed that peripheral nerve injury-induced changes in microglia differ significantly in acute and chronic phases of neuropathic pain. We detected sex-specific changes in gene expression, mainly in the acute phase, and identified a subpopulation of microglia that is selectively induced in males, but not in females, 3 days after peripheral nerve injury. We identified *Apoe* as a top upregulated gene in microglia in chronic phases of neuropathic pain in mice. In humans, we show that polymorphisms in the *APOE* gene are correlated with distinct chronic pain conditions. In addition, we profile microglia from the human spinal cord and reveal the presence of cells with disease-related transcriptional signatures. Collectively, our studies identify the dynamic response of microglia to peripheral nerve injury in mice, link ApoE to chronic pain in mice and humans, and provide the single-cell characterization of human spinal cord microglia.

## Results

**Microglia single-cell transcriptional landscape in neuropathic pain.** Recent single-cell RNA sequencing (scRNA-seq) studies of microglia have revealed the vast heterogeneity of microglia transcriptional states during physiological and pathological conditions, identifying distinct signatures of transcriptional responses to acute injury and chronic disease states such as AD[17,18,21]. To reveal the heterogeneity of microglia at the single-cell level in the development and maintenance of chronic pain, we studied microglia in the spinal cord of mice subjected to the spared nerve injury (SNI) assay of neuropathic pain[22,23] (Fig. 1a). SNI is a common assay of peripheral nerve injury-induced pain, featuring mechanical hypersensitivity that persists for months in both male and female mice (Fig. 1b, c). We collected and analyzed single microglia cells from the lumbar spinal cord of male and female naive mice as well as mice of both sexes at day 3, day 14 and 5 months post-SNI (bilateral) and their age- and sex-matched sham controls (Fig. 1d). Day 3 time point represents an early (acute) pain development phase and corresponds to the peak of microglia proliferation[1]. At day 14, hypersensitivity is fully established, representing a sub-chronic phase of pain development. The 5-month time point was selected to study the chronic phase of neuropathic pain. Microglia from the lumbar spinal cord of naive, sham surgery, and SNI mice (2 replicates per condition, 4 mice pooled per replicate, SNI and sham surgeries performed bilaterally to avoid dissection of the ipsilateral side) were isolated using FACS (Supplementary Fig. 1), gating for CD45$^{low}$CD11B$^{high}$CX3CR1$^{high}$ cells. Purified microglia were processed using the droplet-based 10X Genomics Chromium system, generating high-quality single-cell transcriptomic data (Supplementary Fig. 2a–f) from 188,787 sequenced microglia (~13,500 cells per condition after quality control) with a median number of ~1900 genes per cell at an average sequencing depth of 43,000 reads/cell. Unsupervised clustering analysis of all cells under all conditions revealed 11 distinct clusters (Fig. 1e and Supplementary Figs. 3 and 4). Cells in all clusters expressed canonical microglia genes such as *Tmem119, Fcrls, P2ry12, Cx3cr1, Trem2,* and *C1qa* (Fig. 1f).

**Heterogeneity of spinal microglial transcriptional states.** Gene expression analysis identified genes uniquely expressed in microglia in clusters 4, 5, 6, 7, 8, 10, and 11 (UMAP plot of the top unique gene for each cluster is shown in Fig. 1g and a complete list of uniquely expressed genes in Supplementary Data 1). Microglia in clusters 1, 2, 3, and 9 could not be defined by a single unique gene marker. Gene expression analysis also generated a list of genes with the highest expression level in each microglia cluster (heatmap of the 8 top genes in each cluster is shown in Supplementary Fig. 5 and a complete list in Supplementary Data 2). Clusters 1–6 grouped together within the UMAP plot (Fig. 1e). Cells in these clusters account for the majority of microglia in naive mice (~98%, inset in Fig. 1e). Clusters 7 and 8, together containing ~1% of all microglia in naive mice, showed proximity in the UMAP plot. High expression of cell cycle-associated markers *Mki67* and *Top2a* in microglia in these but not in other clusters (Supplementary Fig. 6a) suggests that cells in clusters 7 and 8 are proliferating microglia. Gene Ontology (GO) analysis of cluster markers showed that microglia in clusters 7 and 8 are enriched for DNA replication and cell cycle genes (Supplementary Fig. 6b). Microglia in cluster 9 exhibit a transcriptional signature characterized by increased interferon and cytokine-mediated signaling (Supplementary Fig. 6c). Cluster 9 microglia were absent in naive mice (0.035% of total), but their number was substantially increased 3 days post-SNI in males (4.9%) but not in females (0.03%) (Fig. 2a–d). Microglia belonging to cluster 9 were largely undetected at day 14 and 5 months post-SNI in both sexes (Fig. 2a, c), demonstrating their nerve injury-induced acute onset and male-specific origin. Microglia in cluster 10 uniquely express *Cldn5* that encodes for the tight junction protein, claudin-5. Claudin-5 is

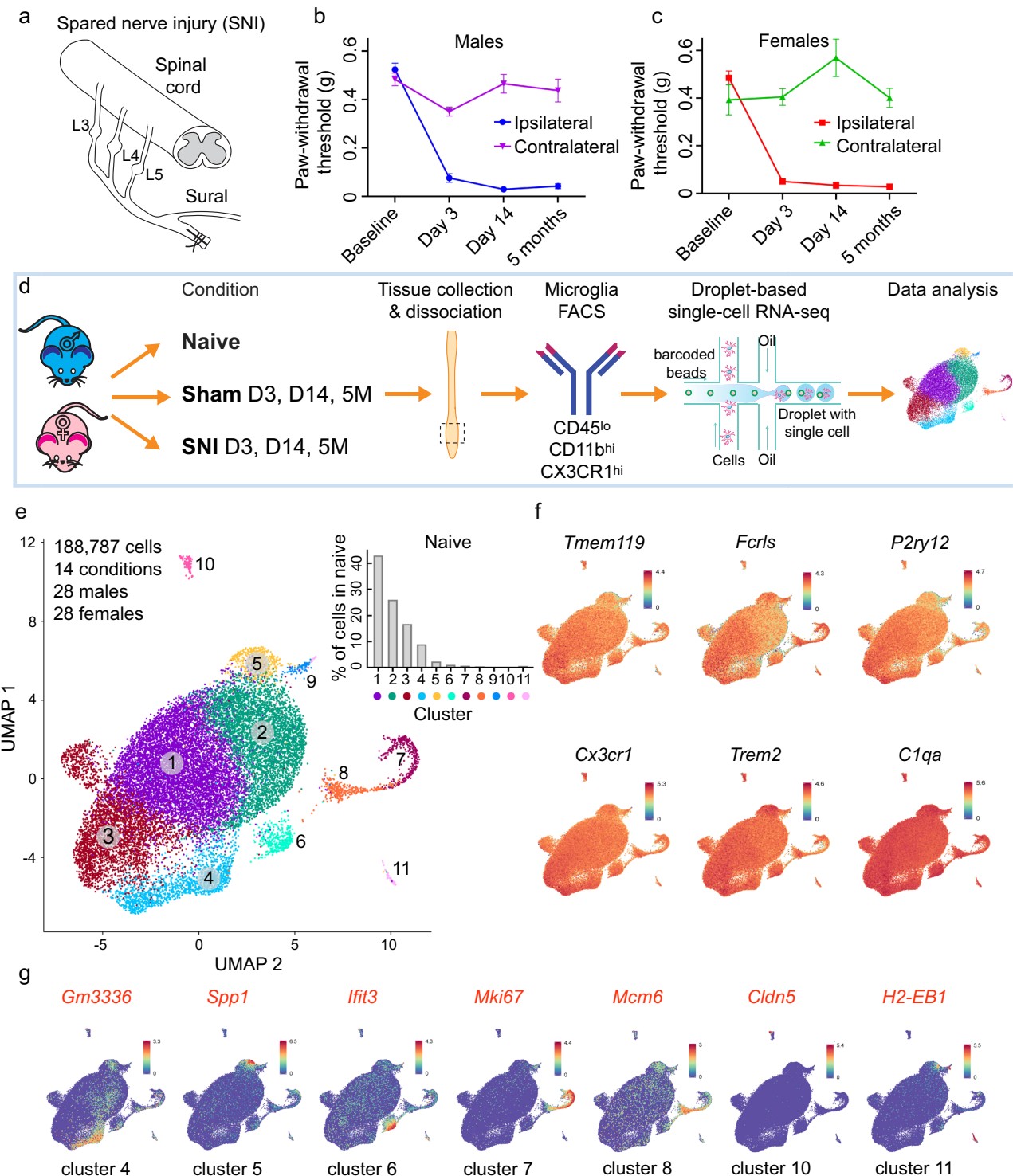

**Fig. 1 Spinal cord microglia are present in several distinct subpopulations. a** A schematic illustration of the spared nerve injury (SNI) model of neuropathic pain. Paw-withdrawal threshold (data are presented as mean ± s.e.m.), as measured by von Frey filaments, was decreased following SNI in male (**b**, *n* = 12/group) and female (**c**, *n* = 8/group) C57BL/6 mice. **d** Microglia were isolated from the lumbar spinal cord of male and female mice, followed by tissue dissociation, FACS purification, and single-cell RNA sequencing (scRNA-seq). **e** UMAP plot reveals that microglia in the mouse spinal cord in all conditions are present in 11 distinct clusters. Inset shows proportion of cells in each cluster in naive mice. **f** UMAP plot showing the expression (log-normalized counts) of canonical microglial genes. **g** Expression of a top unique gene in the indicated cluster and its UMAP plot are shown.

expressed by endothelial cells and also in vessel-associated microglia creating tight junctions with endothelial cells to maintain the blood-brain barrier[24]. In addition to *Cldn5*, cells in cluster 10 express canonical microglia genes, suggesting that this small population (0.19% of total) represents vessel-associated

microglia. Cluster 11 (0.58% of total) was composed of cells expressing canonical microglia genes (*Fcrls, Trem2, Cx3cr1, Tmem119, C1qa,* and *P2ry12*) along with unique expression of macrophage/monocyte markers (*H2-Aa, Mrc1, Ccr2, Lyve1, Dab2, Mgl2, F13a1*) (Supplementary Data 1 and Supplementary

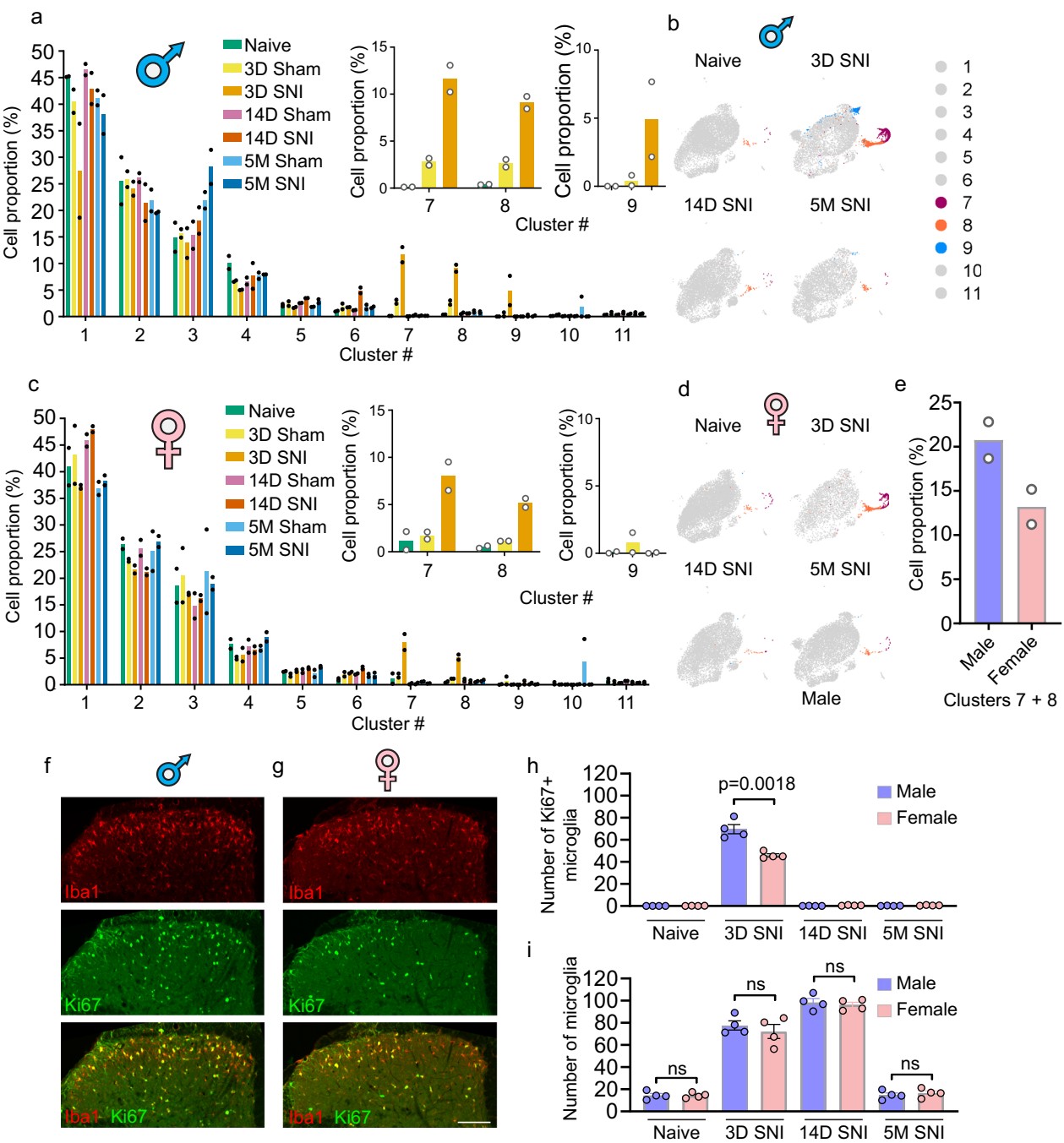

**Fig. 2 Peripheral nerve injury induces changes in proportions of specific microglia subpopulations.** Proportion of microglia in each cluster in males (**a**) and females (**c**) is shown. Insets show the proportions of cells in clusters 7, 8, and 9. UMAPs of indicated conditions in males (**b**) and females (**d**), highlighting in color cluster 7, 8, and 9. **e** Proportion of microglia in clusters 7 and 8, combined, at day 3 post-SNI. Iba1 (microglia, red) and Ki67 (proliferating cells, green) immunostaining reveal proliferating microglia at day 3 post-SNI in males (**f**) and females (**g**). Similar results were obtained in two independent experiments. **h** Quantification of proliferating microglia ($n = 4$ mice/group, 3D SNI: male versus female, $t(6) = 5.342$, $p = 0.0018$, unpaired two-tailed $t$-test). **i** Quantification of total number of microglia ($n = 4$ mice/group, ns (non-significant) indicates $p > 0.05$, unpaired two-tailed $t$-test). Data are plotted as mean ± s.e.m. Scale bar is 100 μm.

Fig. 6d). This subpopulation of cells might be composed of perivascular macrophages that express both macrophage and microglia genes. Perivascular macrophages, which are resident brain macrophages, originate from the same pool of yolk sac hematopoietic progenitors, migrate to the brain early in development, and self-renew, similar to microglia[25–28]. We next employed in situ hybridization (using RNAscope) to detect microglia expressing unique gene markers. Microglia were genetically labeled by expressing tdTomato under the microglia-specific promoter, TMEM119 (tdTomato;TMEM119[CreERT2]). In the dorsal horn of the spinal cord, we detected cells belonging to cluster 4 (*Gm3336*[+]), 5 (*Spp1*[+]), 6 (*Ifit3*[+]), 7 (*Mki67*[+]), 8 (*Mcm6*[+]), 10 (*Cldn5*[+]), and 11 (*H2-EB1*[+]) (Supplementary Fig. 7). Cluster 9 microglia can not be defined by one unique gene marker; however, a fraction of microglia in this cluster (Supplementary Fig. 7f) can be detected by the combination of two

transcripts: the presence of *Lgals1* and absence of *Top2a*. In situ hybridization showed that cluster 9 microglia (*Lgals1*+*Top2a*−) were present in the dorsal horn of male but not female mice 3 days post-SNI (Supplementary Fig. 7g-i). Altogether, our data show that the majority of microglia in the naive spinal cord exist in six transcriptional states (clusters 1–6), in addition to small populations (<2% of total) of proliferating (cluster 7 and 8) and vessel-associated (cluster 10) microglia.

**Peripheral nerve injury induces greater proliferation of microglia in males than in females**. Three days following SNI, the number of microglia in clusters 7 and 8 increased dramatically from 0.5% in male naive mice (5.4% in sham) to 20.8% of all microglia in SNI mice (Fig. 2a, b). Surprisingly, the increase in females was smaller; from 1.7% in naive mice (2.9% in sham) to 13.2% in female SNI mice (Fig. 2c–e), suggesting attenuated proliferation of microglia in females. To corroborate this finding, we labeled proliferating microglia using a marker of proliferating cells, Ki67, co-stained with the microglia marker, Iba1. Quantification of Ki67-positive microglia in the spinal cord dorsal horn (Fig. 2f–h) showed that the number of proliferating microglia was significantly higher in males as compared to females at day 3 post-SNI (Ki67-positive microglia per section; males: 68.8 ± 5.9; females: 46.6 ± 1.8, $p = 0.011$, Fig. 2h), confirming the scRNA-seq result. Greater proliferation of microglia in males than in females at day 3 post-SNI was recapitulated in tdTomato;TMEM119^CreERT2 mice (Supplementary Fig. 8a). Notably, microgliosis (total number of microglia) was unchanged between male and female mice (dorsal horn microglia per section, males: 82.8 ± 5.5; females: 76.4 ± 1.8, $p = 0.31$, Fig. 2i and Supplementary Fig. 8a), suggesting that other mechanisms compensate for reduced proliferation in females.

**Peripheral nerve injury induces profound sex-specific changes in the microglia transcriptome**. To study transcriptional changes in microglia following peripheral nerve injury, we identified differentially expressed genes (DEGs) in SNI versus corresponding sham groups in each microglia cluster at three-time points in male and female mice (Fig. 3a–e and Supplementary Data 3). We detected a large number of DEGs in male and female microglia at day 3 post-SNI; this number decreased substantially on day 14 and 5 months in both sexes (Fig. 3b). Analyses of DEGs in each cluster showed that the vast majority of changes in gene expression occurred in microglia from clusters 1–9 at day 3 post-SNI (Fig. 3c–e) and in clusters 1–6 at later time points.

Interestingly, we observed changes in the expression of numerous mRNAs encoding ribosomal proteins (small ribosomal subunit (Rps) and large ribosomal subunit (Rpl)) in microglia following SNI, an effect that was significantly more pronounced in males than females (Fig. 3f and Supplementary Fig. 8b). Increased expression of ribosomal proteins might promote ribosomal biogenesis and protein synthesis in activated microglia. To obtain insights into the biological processes induced in spinal cord microglia by peripheral nerve injury, we performed GO analysis on DEGs (after removing Rp) in each cluster (Supplementary Data 4) and in clusters 1–6 combined (Fig. 3g). We found that in males, nerve injury induces a strong immune response at day 3 (major GO category: cellular response to cytokine stimulus, inflammatory response, apoptotic process), whereas in females inflammatory processes were less pronounced and top categories included ATF-6-mediated unfolded protein response and receptor-mediated endocytosis (Fig. 3g). At day 14 in males, mitochondrial ATP production and other metabolic processes, including lipid metabolism and trafficking, were the top categories, whereas genes involved in the regulation of

endocytosis/phagocytosis, cell adhesion, and lipid catabolism were affected in females. At 5 months post-SNI, male microglia showed changes in genes related to mitogen-activated protein kinase (MAPK) signaling, whereas female microglia exhibited changes in metabolic processes and responses to lipopolysaccharide (LPS). Microglia in male-specific cluster 9 at day 3 post-SNI exhibited strong inflammatory responses and showed changes in genes related to mitochondrial ATP production and phagocytosis (Fig. 3h).

Sexual dimorphism of microglia involvement in neuropathic pain has been extensively studied, revealing that microglia are necessary for the development of hypersensitivity in males but not in females[3,10,11]. To study the differences in responses of male and female microglia to peripheral nerve injury, we compared DEGs between both sexes. Microglia in males showed more DEGs than females at day 3 post-SNI (Fig. 3a, b), and only a fraction of DEGs was shared between the sexes at each time point (Fig. 3i). Surprisingly, correlational analysis between DEGs in male and female microglia in each of the first six clusters (1–6) on day 3 revealed no or negative correlations (Supplementary Fig. 9a), indicating highly different acute transcriptional responses in male and female microglia to peripheral nerve injury. At later time points (day 14 and 5 months), changes in DEGs in males and females by cluster showed weak to moderately positive correlations (Pearson correlations ranging from $r = 0.12$–$0.55$ on day 14 and from $r = 0.09$–$0.45$ at 5 months post-SNI, Supplementary Fig. 9b, c). Interestingly, we found that MAPK-specific phosphatases *Dusp1* and *Dusp6* are downregulated in clusters 3, 8, and 9 at early (day 3) but not late (day 14 or 5 months) time points post-SNI selectively in male but not female microglia (Supplementary Fig. 10). This finding is consistent with male-specific upregulation of p38 phosphorylation[15] which is central to the activation of microglia following peripheral nerve injury[16,29]. *Il1b* and *Tnf* also showed male-specific upregulation at day 3 post-SNI, whereas *Tspo* was increased in both sexes (Supplementary Fig. 10).

Previous studies have identified several populations of microglia exhibiting unique transcriptional responses to pathological and context-specific physiological conditions. Therefore, we compared (using Fisher's exact test) SNI-induced changes in the microglia transcriptome to previously described transcriptional signatures (Fig. 3j). IRM were identified in the subcortical white matter in response to lysolecithin (LPC)-induced demyelination, a common model of multiple sclerosis (MS)[17]. DAM were found in the brain of a mouse model of AD. Axon tract-associated microglia (ATM) were identified as a transient population detected in the pre-myelinated developing brain in the axon tracts (corpus callosum)[17]. Our analysis revealed that the transcriptional landscape in male microglia in clusters 1–6 was similar to IRM at all time points post-SNI (day 3, day 14 and 5 months) and showed mild similarities to DAM and ATM at day 14 (Fig. 3j and Supplementary Data 5). In contrast, changes in gene expression in female microglia were less similar to the assessed transcriptional signatures, exhibiting similarities to IRM, DAM, and ATM microglia mainly at day 14 post-SNI in clusters 1, 3, 4, 5, and 6 (Fig. 3j). Interestingly, microglia in male-specific cluster 9 at day 3 post-SNI exhibited a transcriptional profile which was highly similar to IRM as compared to other clusters (Fisher's exact test: $p = 5.8 \times 10^{-68}$, Fig. 3k and Supplementary Data 5).

**ApoE is upregulated in microglia in chronic phases of neuropathic pain**. One of the top genes in both the IRM and DAM transcriptional signatures encodes for apolipoprotein E (ApoE), which is involved in lipid metabolism and trafficking and has

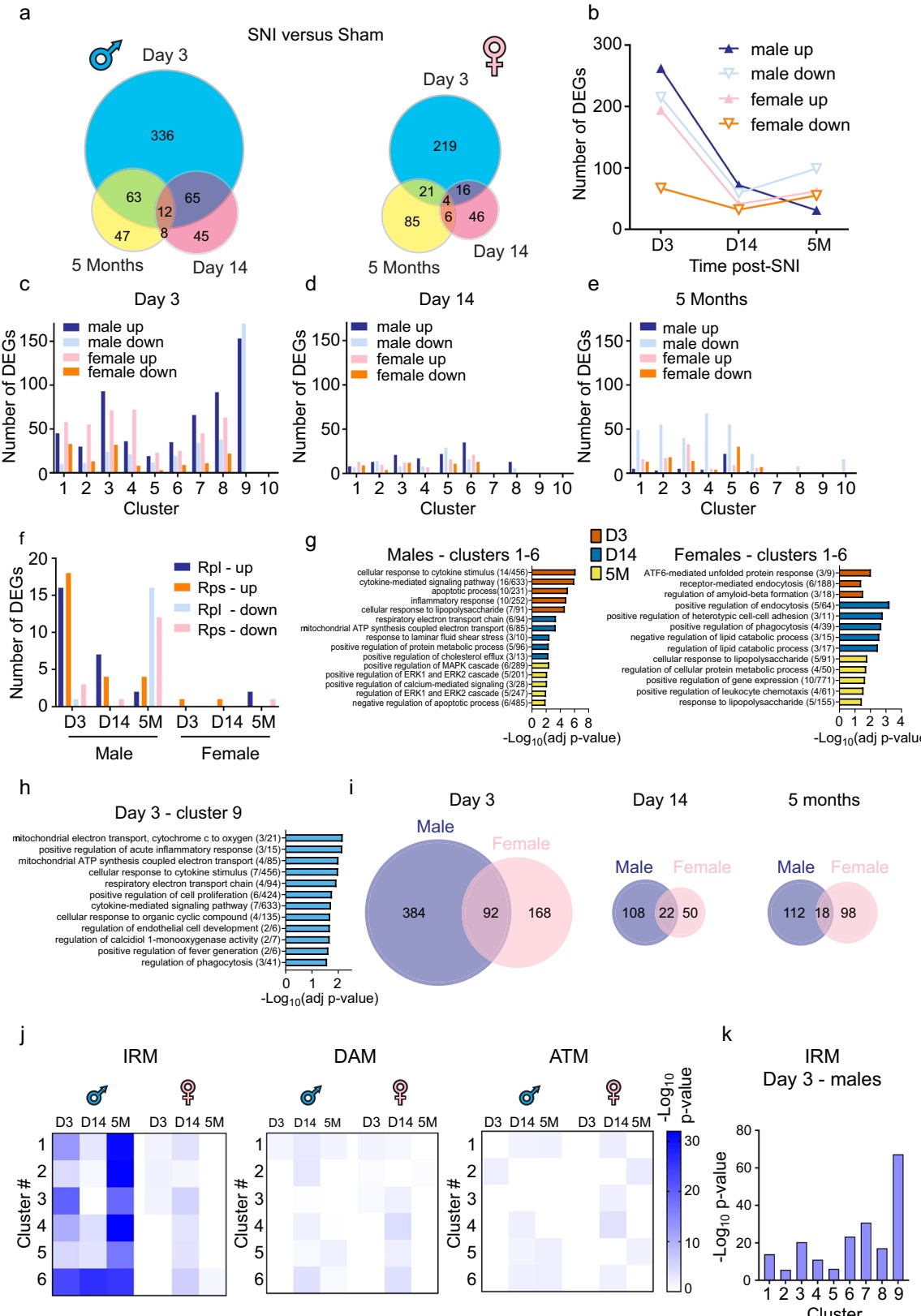

been implicated in AD pathophysiology[30]. Our analyses revealed that *Apoe* is the most abundant transcript in cluster 5 spinal cord microglia in both sexes (Fig. 4a and Supplementary Data 2). Following SNI, *Apoe* mRNA levels in males and females showed no change in expression at day 3 after nerve injury but were significantly upregulated at day 14 (males: top upregulated gene

in clusters 1 and 3 and significantly upregulated in clusters 2, 4, and 6; females: top upregulated gene in clusters 1–6) and 5 months (top upregulated gene in clusters 1–6 in males and females) (Fig. 4a–c). ApoE protein in the spinal cord dorsal horn is present in neurons, astrocytes, and microglia (Fig. 4d). As expected, ApoE immunoreactivity in microglia was found in the

**Fig. 3 Changes in gene expression in microglia after peripheral nerve injury. a** Number of differentially expressed genes (DEGs) at day 3, day 14 and 5 months in males (left) and females (right). **b** Number of upregulated and downregulated DEGs in each sex/direction condition. Number of DEGs per cluster at day 3 (**c**), day 14 (**d**), and 5 months (**e**) post-SNI. **f** Number of upregulated and downregulated mRNAs encoding large (Rpl) and small (Rps) ribosomal protein subunits is shown. **g** Gene Ontology (GO) analysis of DEGs (after removing ribosomal mRNAs) in males and females in each time point for clusters 1-6 combined. The top 20 upregulated DEGs from each cluster were used for the analysis. **h** GO analysis for cluster 9 at day 3 post-SNI in males. The top 40 upregulated genes were used for the analysis. **i** Overlap between DEGs in male and female in each condition is shown. See Supplementary Data 3 for complete lists of DEGs per cluster per condition. **j** Genes in Injury-responsive microglia (IRM), Disease-associated microglia (DAM), and Axon tract-associated microglia (ATM) transcriptional signatures (lists of genes taken from[17], full lists are in Supplementary Data 5) were compared with DEGs in each condition for cluster 1–6 using two-sided Fisher's exact test. Heatmap showing the enrichment for IRM, DAM, and ATM gene expression signature in each condition for cluster 1–6. White indicates no significant overlap ($p > 0.05$). Shades of blue (for $p$ value < 0.05) indicate the significance of the overlap between two gene lists. **k** Enrichment for IRM signature in cluster 1–9 at day 3 post-SNI in males (Fisher's exact test). See Supplementary Data 5 for numerical values of the analysis.

cytoplasm, exhibiting a granular pattern (Fig. 4e). To study if upregulation of *Apoe* transcript in microglia leads to changes in ApoE protein levels, we quantified ApoE in the spinal cord microglia after SNI in both male and female mice. ApoE levels in Iba1-labeled microglia were low at day 3 post-SNI, but increased significantly at day 14 and 5 months (Fig. 4f, g). These findings demonstrate that ApoE is persistently upregulated in microglia following peripheral nerve injury and suggest that ApoE might have a role in chronic responses of microglia to peripheral nerve injury.

**Polymorphisms in *APOE* gene in humans is associated with chronic pain.** Persistent upregulation of microglial ApoE after peripheral nerve injury in mice prompted us to study the link between *APOE* and chronic pain in humans. Polymorphisms in *APOE* are associated with several chronic diseases of the CNS. For example, it represents a major genetic risk factor for late-onset AD[30]. *APOE* has 3 major haplotype-based allelic variants: ε2, ε3, and ε4 (Fig. 5a). *APOE-ε4* confers increased risk[31], whereas *APOE-ε2* confers decreased risk[32] for AD, as compared to the common *APOE-ε3* allele. We assessed the association of *APOE-ε2* and *APOE-ε4* variants with reports of pain at several body sites (Fig. 5b) in males, females, and combined, in the large UK Biobank cohort (Fig. 5c–e). Strikingly, we found that *APOE* polymorphisms were associated with distinct chronic pain reports (Fig. 5c–e and Supplementary Data 6). Male carriers of *APOE-ε4* have reduced risk to develop chronic back pain (odds ratio (OR) = 0.41, $p = 0.0001$), and headaches (OR = 0.67, $p = 0.018$) (Fig. 5c). Conversely, *APOE-ε2* in males confers increased risk to develop chronic back pain (OR = 1.96, $p = 0.0001$), hip pain (odds ratio (OR) = 1.35, $p = 0.022$), knee pain (OR = 1.63, $p = 0.005$), and stomach pain (odds ratio (OR) = 1.25, $p = 0.036$), and slightly decreased risk to develop chronic facial pain (OR = 0.91, $p = 0.029$) (Fig. 5c). For acute pain, only *APOE-ε2* carriers have an increased risk to develop headaches (OR = 1.35, $p = 0.046$). In females, the association between *APOE* polymorphisms and pain was substantially weaker and found only for carriers of *APOE-ε4* who have reduced risk to develop chronic back pain (OR = 0.45, $p = 0.0007$), and headaches (OR = 0.65, $p = 0.0045$) (Fig. 5d and Supplementary Data 6). Collectively, our data indicate that, in direct opposition to AD, carriers of *APOE-ε4* have a decreased risk whereas carriers of *APOE-ε2* have an increased risk to develop distinct chronic pain conditions.

**Characterization of human spinal cord microglia at the single-cell level.** To elucidate the role of spinal cord microglia in chronic pain and other spinal cord pathologies, such as MS and amyotrophic lateral sclerosis (ALS)[33], and translate this knowledge to humans, it is imperative to study human spinal cord microglia heterogeneity and transcriptional states. Accordingly, we

investigated microglia at the single-cell level in human spinal cord samples from two females (53- and 68-years old) and one male (39-years old). Spinal cord tissues were obtained through a rapid autopsy organ donor program from individuals without documented history of chronic pain or neurological disease. Spinal cord tissue was dissociated and processed using droplet-based scRNA-seq, followed by isolation of microglia in silico (expressing either *TMEM119*, *ITGAM* (*CD11b* in mouse) or *CX3CR1*), resulting in high-quality transcriptional profiles from 3823 cells (Supplementary Fig. 11a, b). Unsupervised clustering analysis identified eight distinct clusters of microglia (Fig. 6a, b, Supplementary Data 7), all expressing canonical microglia markers *TREM2* and *C1QA*. In addition to the main population of microglia, we observed a cluster of cells expressing mixed microglia-oligodendrocyte genes (cluster 9) and a minor population of T/NK cells (cluster 10). Cluster 1 was the largest microglia cluster, containing >15% of total cells analyzed. This cluster was characterised by a high expression of genes associated with 'homeostatic' microglia[34] such as *CSF1R*, *BHBLE41*, *FCGR1A*, *CTSS*, and *GPR34*. Microglia in cluster 3 were characterized by the expression of chemokines *CCL3* and *CCL4*, and the zinc finger containing transcription factors early growth response protein 1 and 2 (EGR1 and EGR2). A recent study of 1180 human cortical microglia derived from surgically resected temporal lobe tissue identified a similar cluster of cells characterized by high expression of *CCL2*, *CCL4*, *EGR2*, and *EGR3*, with the authors denoting these cells as 'pre-activated'[19]. Prior to attaining a mature surveillant phenotype, pre-activated microglia express high levels of *EGR1* and are closely associated with synaptic pruning and neural maturation[35]. Microglia in cluster 3 also expressed high levels of complement component C3 which is associated with synapse elimination and the development of precise synaptic connectivity[36,37]. Collectively, the profile of spinal cord microglia in cluster 3 was that of cells interacting with and potentially modulating neuronal structure and function, as suggested by associated GO terms (Fig. 6c). Interestingly, gene expression in cluster 4 microglia overlapped with the IRM/DAM/ATM-like signatures (Fig. 6d and Supplementary Data 8), with a shared common transcriptional signature of 5 core genes: *LPL*, *SPP1*, *APOE*, *CD63*, and *CTSB*. GO terms associated with cluster 6 highlight functions of 'translation', 'peptide biosynthetic process', and 'translational initiation', suggesting that cells in this cluster were actively generating new proteins (Fig. 6c). High expression of immunoproteasome subunit genes (*PSMA3/A4/A5/A7*, *PSME2*, *PSMB2/B3/B6/B8/B9*, and *PSMD9*) identify microglia in cluster 7 as actively processing antigen for presentation on MHC-I[38]. Intriguingly, cluster 7 microglia also showed high expression of genes involved in the pathogenesis of Parkinson's disease, such as synuclein and protein deglycase DJ-1 (*SNCA* and *PARK7*). Finally, cluster 7 microglia exhibited high expression of *TSPO*, also shown to be upregulated in mouse microglia at day 3

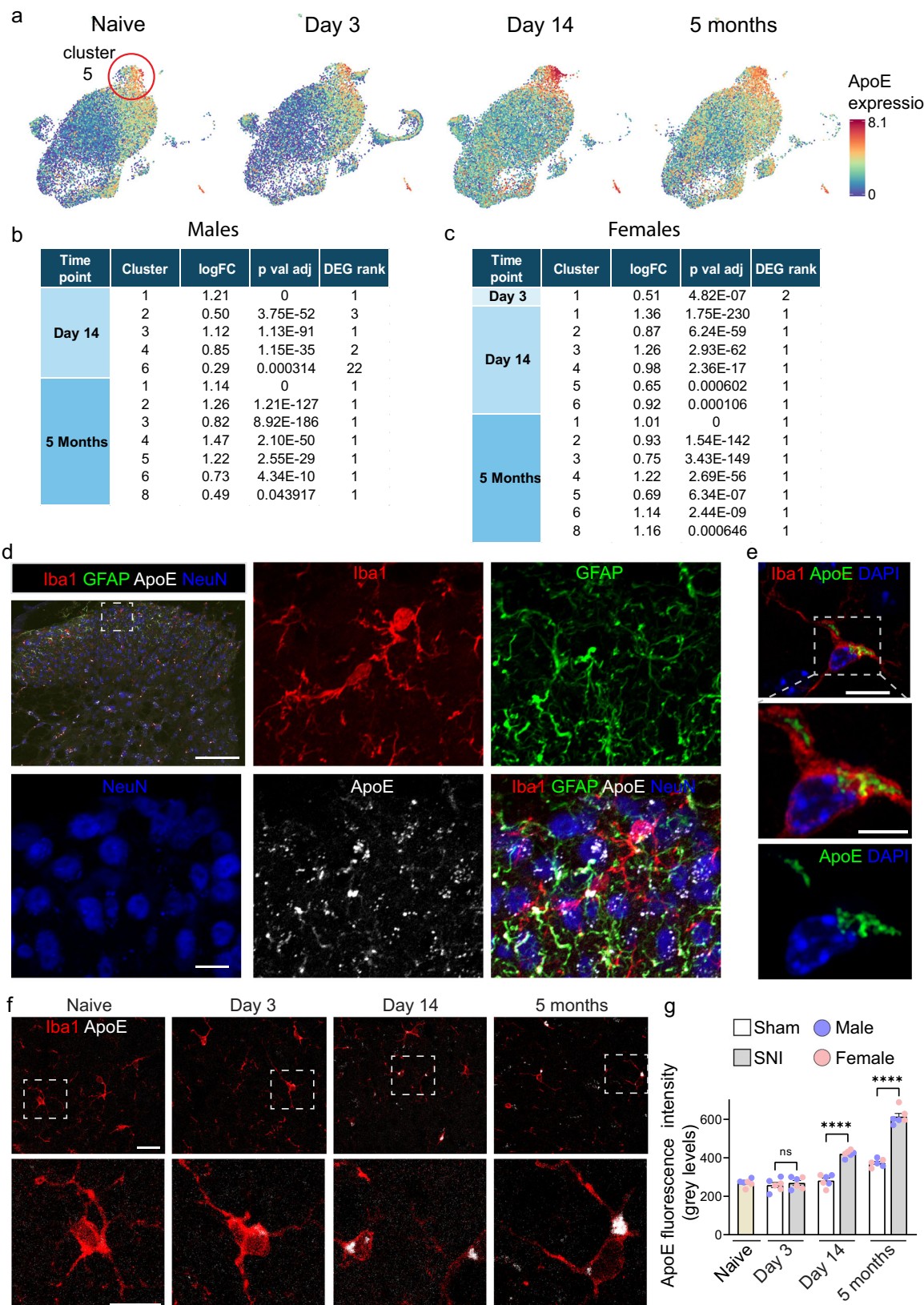

Figure content — Males and Females tables:

**Males**

| Time point | Cluster | logFC | p val adj | DEG rank |
|---|---|---|---|---|
| Day 14 | 1 | 1.21 | 0 | 1 |
| | 2 | 0.50 | 3.75E-52 | 3 |
| | 3 | 1.12 | 1.13E-91 | 1 |
| | 4 | 0.85 | 1.15E-35 | 2 |
| | 6 | 0.29 | 0.000314 | 22 |
| 5 Months | 1 | 1.14 | 0 | 1 |
| | 2 | 1.26 | 1.21E-127 | 1 |
| | 3 | 0.82 | 8.92E-186 | 1 |
| | 4 | 1.47 | 2.10E-50 | 1 |
| | 5 | 1.22 | 2.55E-29 | 1 |
| | 6 | 0.73 | 4.34E-10 | 1 |
| | 8 | 0.49 | 0.043917 | 1 |

**Females**

| Time point | Cluster | logFC | p val adj | DEG rank |
|---|---|---|---|---|
| Day 3 | 1 | 0.51 | 4.82E-07 | 2 |
| Day 14 | 1 | 1.36 | 1.75E-230 | 1 |
| | 2 | 0.87 | 6.24E-59 | 1 |
| | 3 | 1.26 | 2.93E-62 | 1 |
| | 4 | 0.98 | 2.36E-17 | 1 |
| | 5 | 0.65 | 0.000602 | 1 |
| | 6 | 0.92 | 0.000106 | 1 |
| 5 Months | 1 | 1.01 | 0 | 1 |
| | 2 | 0.93 | 1.54E-142 | 1 |
| | 3 | 0.75 | 3.43E-149 | 1 |
| | 4 | 1.22 | 2.69E-56 | 1 |
| | 5 | 0.69 | 6.34E-07 | 1 |
| | 6 | 1.14 | 2.44E-09 | 1 |
| | 8 | 1.16 | 0.000646 | 1 |

post-SNI. GO analysis of DEGs in cluster 8 microglia suggested that cells in this cluster were primarily undergoing cell division. The most significant terms for cluster 8 include 'mitotic cell cycle', 'cell division', and 'cell cycle phase transition' (Fig. 6c). Cells in cluster 2 had acquired a transcriptomic signature dominated by immediate early genes, likely induced by the cellular

dissociation process. This gene set has previously been identified in murine single-cell sequencing studies and includes *DNAJB1, HSPA1A, DUSP1, FOS, FOSB, JUN, HSPA1B, HSP90AA1, HSP90AB1, HSP90B1, HSPA8,* and *HSPB1*[39]. These cells cluster together and make up ~15% of the total dataset, whereas all other clusters are relatively unaffected by the processing. Finally, we

**Fig. 4 ApoE is increased in microglia in chronic phases of neuropathic pain. a** UMAPS of *Apoe* in different conditions in male mice. Color codes for *Apoe* log-normalized counts. Tables show fold change (LogFC) and rank of *Apoe* mRNA in the list of DEGs in males (**b**) and females (**c**). Cluster markers were identified using the *FindallMakers* function in Seurat (two-sided Wilcoxon rank-sum test with Bonferroni correction. Differentially expressed genes between groups were calculated using the Wilcoxon rank-sum test (two-sided) with Bonferroni correction. **d** Low magnification (top left) and high magnification images of the marked area showing immunostaining against Iba1, GFAP, ApoE, and NeuN in the mouse dorsal horn spinal cord section at day 14 post-SNI. Scale bar is 100 μm for low magnification and 10 μm for high magnification images. **e** Representative Airyscan images of Iba1, ApoE, and DAPI in the mouse dorsal horn, showing cytoplasmatic expression of ApoE. Scale bar is 10 μm for low magnification and 5 μm for high magnification images. **f** Representative images of immunostaining for ApoE and Iba1 in the spinal cord of male mice after SNI at day 3, day 14 and 5 months. Bottom images are magnification of area marked in corresponding upper images. Scale bar is 20 μm for low magnification and 10 μm for high magnification images. Similar results were obtained in two independent experiments. **g** Quantification of ApoE immunostaining signal in microglia in males and females ($n = 3$ males and three females per condition). ApoE was increased at day 14 and 5 months post-SNI as compared to corresponding sham groups (Day 3, sham versus SNI, $q(35) = 0.917$, $p = 0.99$; Day 14, sham versus SNI, $q(35) = 11.93$, $p < 0.0001$; 5 Months, sham versus SNI, $q(35) = 20.58$, $p < 0.0001$. Data are plotted as mean ± s.e.m., one-way ANOVA followed by Tukey's multiple comparisons *post hoc* test. ****$p < 0.0001$; ns, not significant.

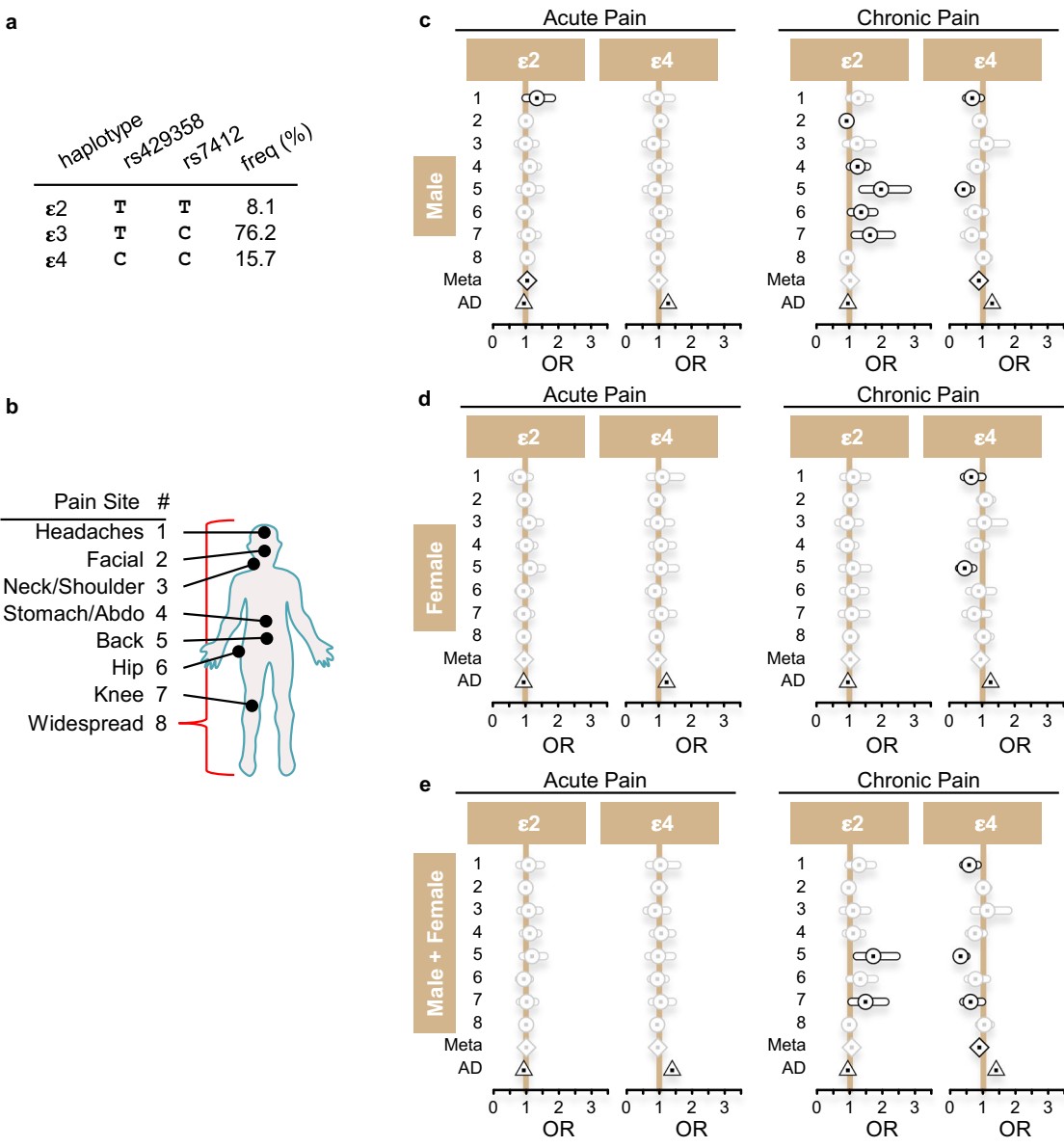

**Fig. 5 Polymorphisms in *APOE* are associated with chronic pain in humans. a** Haplotypes in the human *APOE* gene. The haplotypes are composed of specific alleles of SNPs rs429358 and rs7412, at distinct frequencies in the UK Biobank (UKB) cohort. ε3 is the ancestral haplotype. **b** A schematic diagram of human body indicating reported pain sites. **c** Haplotypic effects of *APOE* in human pain in males. For **c–e**, the effects are depicted on the odds ratios (OR) scale, with 95% confidence intervals, for each pain sites (circles), for inverse standard-error weighted meta-analyzed results (Meta, lozenges), and for Alzheimer's disease (AD, triangle). From left to right; effect of ε2 and ε4 in acute pain, and ε2 and ε4 in chronic pain. Insignificant *P*-values ($P > 0.05$) grayed out. **d** Haplotypic effects in females only. **e** Haplotypic effects in males and females combined. See methods section for description of analyses and statistical approaches. See Supplementary Data 6 for details of the analyses, including statistics and number of cases/controls.

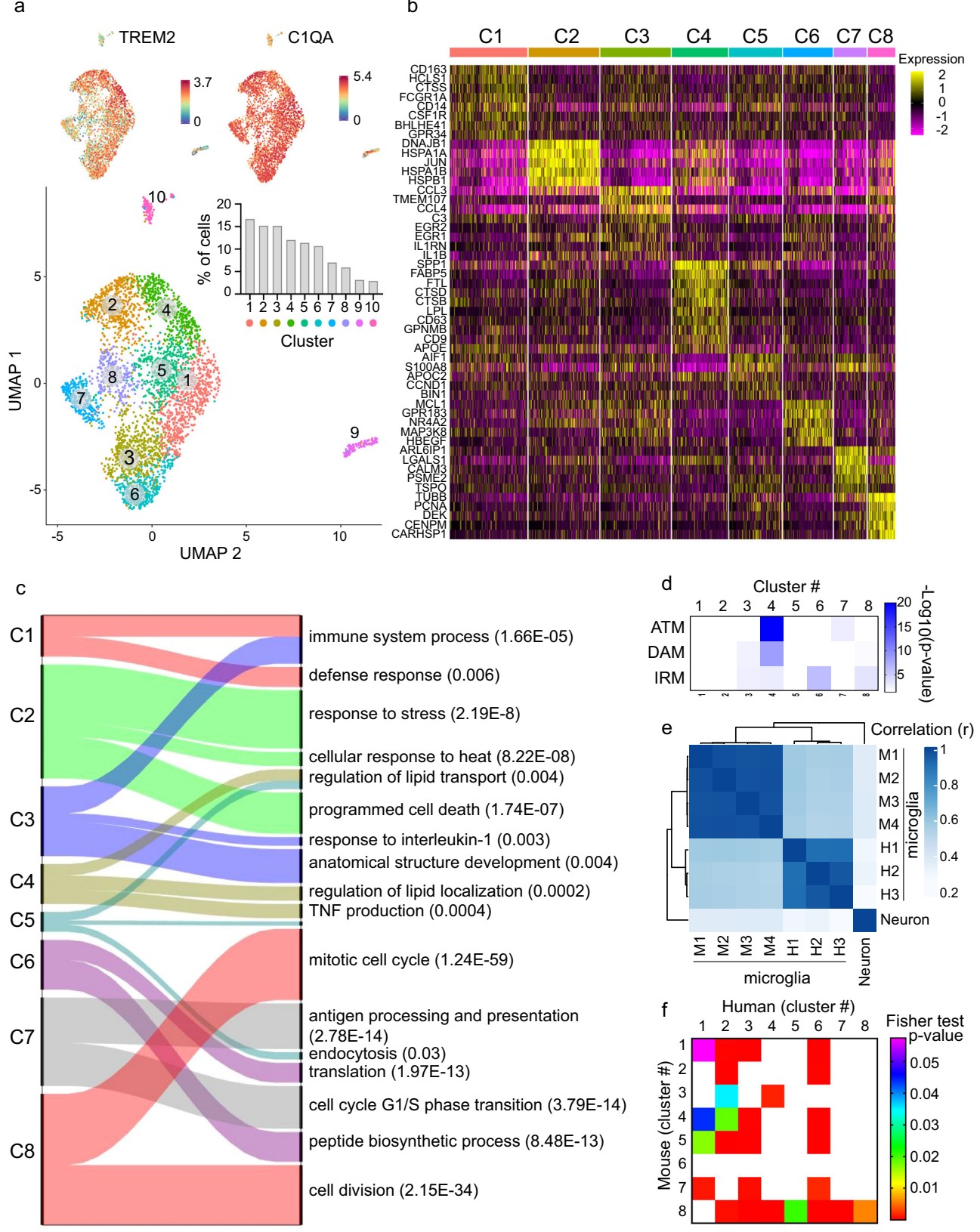

compared the expression signature of human spinal cord microglia with publicly available human non-spinal cord (brain) microglia datasets[19,40,41]. Our analysis suggests that while spinal cord microglia have a common transcriptional landscape with brain microglia, a subset of genes is enriched in spinal cord microglia (Supplementary Fig. 11c–e, and Supplementary

Data 9). In an analysis of common genes upregulated in the spinal cord and brain microglia clusters, we note that spinal cord microglia cluster 3 shows the greatest similarity with brain microglia clusters while little overlap between spinal cord clusters 7 and 8 and brain microglia clusters was observed, suggesting the existence of unique microglia subsets in the human spinal cord

**Fig. 6 scRNA-seq analysis of human spinal cord microglia. a** UMAP plot reveals that cells in the human spinal cord are present in 8 unique clusters (#1–8), plus two additional non-microglia clusters (#9–10). Inset shows the proportion of cells in each cluster. Expression (log-normalized counts) of canonical marker TREM2 and C1QA shown on top. **b** Heatmap showing expression of top 8 highly expressed genes in each cluster. Expression of genes is represented using a z-score value in which yellow indicates higher expression and purple indicates lower expression. See Supplementary Data 7 for a complete list of clusters markers. **c** Alluvial plot depicting the most affected biological processes for each cluster. Upregulated genes of each cluster were used for the analysis. Ribbon thickness indicates the number of genes per biological term. P value for each term is shown in brackets. **d** Heatmap showing the enrichment for IRM, DAM, and ATM gene expression signature in cluster markers for each cluster (two-sided Fisher's exact test). White indicates no significant enrichment ($p > 0.05$). Shades of blue (for p values < 0.05) indicate the significance of the overlap between two gene lists. See Supplementary Data 8 for numerical values and a list of shared genes. **e** Heatmap showing correlation of transcriptomes (treated as bulk) of human microglia (H1, H2, and H3), mouse microglia (M1, M2, M3, and M4) and mouse neurons (neuron dataset taken from ref. [68]). Pearson correlation coefficient (r) is color coded by shades of blue. **f** Analysis of overlap between human and mouse microglia highly expressed transcripts for each cluster (two-sided Fisher's exact test). The color-coded p-values indicate the significance of overlap. White indicates no significant overlap ($p > 0.05$).

(Supplementary Fig. 11f–h and Supplementary Data 9). However, further validation of this spinal cord microglia signature is required.

We observe a strong correlation between human and mouse microglia (Fig. 6e), particularly in human clusters 1–3 and 6 (Fig. 6f). The most striking observation from these data is the identification of cluster 4 which displays remarkable similarity to previously identified transient developmental (ATM) and disease-associated signatures (IRM/DAM) in the mouse brain.

**Association of cluster-specific microglia transcriptomes with human pain genes**. To investigate whether genes in distinct human and mouse microglia clusters are enriched for pain-relevant genes identified in human genome-wide association studies (GWAS), we mapped highly expressed genes in each cluster to GWAS UK Biobank results on several human pain states. This analysis revealed associations between transcriptional signatures of distinct mouse and human microglia clusters to different acute and chronic pain states (Supplementary Fig. 12a–m; Supplementary Data 10). In mice, the strongest association was found for microglia clusters 4 and 9, which showed a strong correlation with GWAS-identified genes associated with acute back pain (Supplementary Fig. 12; Supplementary Data 10), whereas the transcriptome of cluster 2 microglia was associated with genes linked to chronic stomach/abdominal pain. Top individual genes in cluster 2 contributing to the association were *TANK* (tumor necrosis factor receptor-associated factor), which is involved in the regulation of TNF and NF-κB signaling, and *TPM3* (tropomyosin 3), which regulates actin dynamics (Supplementary Fig. 12n and Supplementary Data 11). Human microglia in clusters 2, 6, and 7 were associated with genes related to acute facial, headache, and abdominal pain, respectively. The top contributing genes were *SRC*, a non-receptor protein tyrosine kinase involved in immune response, cell adhesion, and cell cycle progression; *CTSB* (cathepsin B) encoding a member of the C1 family of peptidases; and *SMC4* (Structural Maintenance of Chromosomes 4) involved in chromosome segregation during mitosis. These findings reveal associations between distinct microglia clusters and pain states, supporting the potential role of these subpopulations in mediating pain in humans.

## Discussion

In this study, we analysed the heterogeneity of microglia in the mouse and human spinal cord, and investigated microglial responses to peripheral nerve injury in mice at three time points and both sexes by transcriptionally profiling a large number of microglia using scRNA-seq.

Single-cell transcriptional analyses revealed that mouse and human spinal cord microglia exist in numerous heterogeneous subpopulations. Unsupervised clustering analysis of gene expression revealed that the majority of microglia in mice exist in six clusters (1–6), constituting ~98% of all microglia in naive animals. Small populations (<1% each) of vessel-associated microglia (cluster 10) and perivascular macrophages (cluster 11), which were recently implicated in neuropathic pain[42], were also detected. Peripheral nerve injury resulted in the appearance of three additional clusters: two subtypes of proliferating microglia (clusters 7 and 8) and IRM-like male-specific microglia (cluster 9). In human spinal cord, eight distinct microglia subpopulations were detected, consistent with a recent study showing that human microglia, collected from brain tissue, exhibit greater heterogeneity as compared to other species[18]. Intriguingly, in the human spinal cord, we detected a large subpopulation of microglia (cluster 4) exhibiting a transcriptional profile highly similar to the previously described mouse disease (IRM/DAM)/ early development (ATM) signatures. The identification of IRM/DAM/ATM in tissue isolated from 'healthy' individuals may be indicative of an undiagnosed neurological condition, or suggest that this population, while not observed in the mature mouse brain, may develop in the normally aging human spinal cord.

Consistent with previous studies, we found a robust inflammatory response in male microglia at the acute phase (day 3) following peripheral nerve injury. A large number of genes showed differential expression, many related to immune functions. At day 14 and 5 months post-SNI, the number of DEGs was significantly reduced, and metabolic processes, including energy production and lipid metabolism, were predominantly induced. These data indicate that following the initial immune response to injury, there is a transcriptional shift from immunoreactive microglia towards a metabolically altered state. This phenomenon has been observed in macrophages in response to damage[43] and in microglia in other chronic brain pathologies such as AD and MS[44], revealing that following prolonged activation, microglia shift from mitochondrial oxidative phosphorylation to glycolysis (Warburg effect) for ATP production[45,46], and upregulate genes involved in lipid, cholesterol and lipoprotein metabolism[47].

In females, acute inflammatory responses were less pronounced as compared to males, and predominant processes included endocytosis, cellular metabolism, and cell adhesion. Interestingly, we found that a large number of ribosomal proteins (Rpl and Rps) showed an increased expression selectively in male but not in female microglia at day 3 post-SNI. Induction of ribosomal proteins might enhance the microglia's capacity to produce new proteins, which are required during the phenotypic switch, via increased ribosomal biogenesis and mRNA translation[48].

We found that changes in gene expression in response to peripheral nerve injury differ significantly between males and females at day 3 post-SNI but show similarities at later time

points. Correlation analysis between DEGs in male and female microglia revealed negative or no correlation between males and female at day 3 post-SNI, but weak-to-moderate correlation at day 14 and 5 months. This is consistent with the divergent acute responses of male and female microglia, wherein microglia in males show stronger immune activation as compared to females.

We observed several additional sex differences of note. A subpopulation of microglia (cluster 9) with a marked inflammatory profile was detected exclusively in males at day 3 post-SNI, but not in females, and not at later time points. These microglia, which constitute ~4.5% of all microglia at day 3 in males, show a pronounced IRM signature. We also found that microglia proliferate less in females as compared to males at day 3 post-SNI despite equal microgliosis. Several potential mechanisms, which should be evaluated in future studies, could account for unaltered total microglia count between male and female mice at day 3 post-SNI despite reduced proliferation in females. These mechanisms include: (a) different temporal dynamics of microglia proliferation in males and females, (b) increased apoptosis of male microglia at day 3 post-SNI (supported by altered expression of genes related to apoptosis in male microglia at this time point), and (c) enhanced recruitment of microglia from other areas of the spinal cord in females.

Our scRNA-seq analysis revealed a robust, across-cluster upregulation of *Apoe* mRNA levels in microglia at late (day 14 and 5 months) but not acute (day 3) time points after peripheral nerve injury in both sexes (Fig. 4). *Apoe* was the top upregulated gene in numerous clusters at day 14 and 5 months. Remarkably, we found a strong link between *APOE* and chronic pain in humans by demonstrating that polymorphisms in *APOE* are associated with distinct chronic pain states (Fig. 5). In men, *APOE-ε4* confers a decreased risk to develop chronic headaches and back pain whereas *APOE-ε2* increases the risk to develop chronic abdominal, back, hip, and knee pain. This is opposite to the direction of associations found in individuals with AD, indicating that carriers of *APOE-ε4* have increased risk to develop AD, but decreased risk for developing chronic pain. The association in women was much less pronounced and found only for *APOE-ε4* (decreased risk for chronic headaches and back pain) but not *APOE-ε2*.

ApoE is the most abundant apolipoprotein in the CNS, and is involved in trafficking and metabolism of lipoproteins and cholesterol[30]. ApoE may promote the efflux of intracellular cholesterol which accumulates in microglia following phagocytosis of dying cells and myelin debris. Lipids in microglia are used as precursors of many inflammatory mediators and as a source of energy via oxidative metabolism. Therefore, the regulation of lipid metabolism by ApoE may have a direct effect on microglia inflammatory functions and energy production. Future studies should examine the mechanisms by which different *APOE* isoforms (*APOE-ε2* and *APOE-ε4*) and *APOE*-expressing cell types (microglia, astrocytes, and neurons) modulate the development of chronic hypersensitivity states.

We propose that cluster 10 microglia (0.19% of total microglia count) represent vessel-associated microglia. However, cells in this cluster express several endothelial genes (e.g., *Pecam1*). Thus, we cannot rule out the contamination of this subpopulation by endothelial cells during processing or incomplete dissociation from tightly associated endothelial cells. Alternatively, the presence of endothelial genes could be due to the phagocytosis of endothelial cell fragments by microglia.

In summary, our datasets and analyses represent a comprehensive characterization of spinal cord microglia heterogeneity, providing the basis for elucidating the roles of specific microglia subpopulations and transcriptional states in distinct processes underlying chronic pain and other spinal cord pathologies (e.g.,

MS and ALS). Detection of a male-specific microglia subpopulation (cluster 9) with a marked inflammatory profile might lead to a better understanding of sex-specific mechanisms and the development of targeted therapeutic approaches. The switch from an immune to an altered metabolic state of microglia is a potential mechanism underlying chronic and clinically relevant phases of neuropathic pain. Upregulation of *Apoe*, a gene that we show is strongly associated with human chronic pain conditions, might represent a central mechanism in this switch, and its further examination can generate important insights into the microglia-dependent mechanisms of neuropathic pain.

## Methods

**Animals and environment.** All mouse experiments were approved by the Animal Care Committee at McGill University and complied with Canadian Council on Animal Care guidelines. C57BL/6J mice (8–12 weeks of age) of both sexes were obtained from The Jackson Laboratory (Bar Harbor, ME, USA). Mice were housed at McGill University animal facility in standard shoebox cages (5 per cage) and maintained at a temperature-controlled environment on a 12:12 h light/dark cycle (lights on at 07:00 h). Food (Envigo Teklad 8604, Lachine, QC, Canada) and water were provided ad libitum. Reporter Ai14 tdTomato mice (The Jackson laboratory, stock #007914) were crossed with TMEM119[CreERT2] mice (the Jackson laboratory stock #031820) to generate TdTomato;TMEM119[CreERT2] animals. The experimenter was blinded to the genotype/condition in all studies.

**SNI surgeries.** Mice were deeply anesthetized with isoflurane and subjected to bilateral SNI or sham surgeries. SNI surgery was performed as described previously[22,23]. Briefly, the sciatic nerve was exposed after making an incision on the skin on the lateral surface of the mouse thigh and sectioning through the biceps femoris muscle. Two of the three-terminal branches of the sciatic nerve, the tibial and common peroneal nerves, were tightly ligated with 7.0 silk (Covidien, S-1768K) and 2–4 mm of the nerve distal to the ligation were removed, avoiding any disturbance of the sural nerve. The muscle and skin were closed in separate layers using coated Vicryl (Ethicon, J489G). For the sham surgeries, the sciatic nerve, as well as its three branches, were exposed but left intact.

**Behavioral studies.** All experiments took place during the light cycle, no earlier than 09:00 h and no later than 16:00 h. Mice that underwent unilateral SNI were placed in custom-made Plexiglas cubicles (5.3 × 8.5 × 3.6 cm) on a perforated metal floor and were habituated for at least 1 h before testing. The up-down method of Dixon was used to estimate 50% withdrawal thresholds using calibrated von Frey nylon monofilaments (Stoelting Touch Test). Filaments were applied to the plantar surface of the hind paw for 3 s and responses were recorded. At least 2 consecutive measures were taken on each hind paw at each time point and averaged. Mice were tested for mechanical sensitivity using von Frey fibers at baseline and 3 days, 14 days, and 5 months after surgery to quantify mechanical allodynia.

**Mouse microglia single-cell suspension.** Fresh spinal cord samples (L4-L5 segment) were collected at day 3, day 14 and 5 months post-SNI or sham surgery. One biological sample consisted of lumbar segments pooled from four mice. The microglia single-cell suspension protocol was adapted from[17]. Mice were deeply anesthetized and transcardially perfused with ice-cold Hank's balanced salt solution (HBSS). Then, the spinal cords were extracted using 3 mL of HBSS by inserting a 20 G needle at the caudal end of the vertebral column. A 5 mm section of the spinal cord (lumbar region) was dissected and put into well plates containing 2 mg ml$^{-1}$ collagenase type IV (Gibco) in 3 ml DMEM. Spinal cords were minced using scissors and treated with 1 µl of DNase I (Thermo Scientific) for 45 min under incubation at 37 °C. After 20 mins into the incubation, the tissue was Dounce homogenized 20 times with the pestle A while simultaneously rotating the pestle. The cell suspensions were then transferred back into the well plate and gently agitated by hand 10 min later. Then, the cell suspensions were passed through a pre-wet (with DMEM) 70-µm cell strainer, transferred to a prechilled 50 mL tube and centrifuged for 10 min at 22 °C (400 vcf, acceleration: 2, brake: 1). Samples were washed by gently decanting the supernatant and adding 15–20 mL 1X HBSS followed by centrifugation for 10 min at 22 °C (400 vcf, acceleration: 2, brake: 1). The supernatant was decanted, and the cell pellets re-suspended in 13 mL of 30% Percoll (Sigma) diluted in HBSS at room temperature. The 13 mL of cell suspension were transferred to a 15 mL tube and overlaid with 2 mL of HBSS. Cells were then centrifuged for 20 min at 22 °C (400 vcf, acceleration: 1, brake: 0). The debris and Percoll were carefully removed without disturbing the pellet of immune cells at the bottom of the tube. The cell pellet was washed with 15 mL of ice-cold HBSS and centrifuged for 7 min at 4 °C (500 vcf, acceleration: 2, brake: 1). The HBSS was removed from the tubes and the samples were re-suspended in 500 µl of ice-cold FACS buffer (0.4% BSA (non-acetylated) in 1x PBS, sterile) with CD11b (PE), CD45 (APC-Cy7) and CX3CR1 from Biolegend at a 1:200 dilution for 1 h on ice and protected from light. Samples were washed in 14 mL of ice-cold HBSS and

centrifuged for 7 min at 4 °C (500 vcf, acceleration: 2, brake: 1). Finally, the supernatant was carefully removed, and the pellet was re-suspended in 500 µl of ice-cold FACS buffer. The samples were kept on ice and protected from light until they were ready to sort. Samples were sorted using either a FACSAria III cell sorter equipped with 405-nm, 488-nm and 640-nm lasers and the appropriate filters or a FACSAria Fusion equipped with a 405, 488, 561, and 633-nm lasers and the appropriate filters (both from BD Biosciences). In both cases, live, single CD11b$^{hi+}$, CD45$^{low}$, CX3CR1$^{hi+}$ cells were sorted using a 70-µm nozzle at 70 psi. Gates were determined using fluorescence minus one sample. Single cells were discriminated from aggregates or doublets by gating the Forward Scatter-Width (FSC-W) low events in a Forward Scatter-Height (FSC-H) versus FSC-W bivariate plot. A second screening was then performed by selecting the Side Scatter-Width low (SSC-W) events in a Side Scatter-Height (SSC-H) versus SSC-W bivariate plot.

**Single-cell RNA sequencing.** Around 8700 cells per sample were loaded into a Chromium Single Cell Chip (10x Genomics). Subsequent reverse transcription and library preparation was performed using the Chromium Single Cell 3′ Library & Gel Bead Kit v3 (10X Genomics) following manufacturer instructions. Samples were sequenced to an average depth of 43,000 reads per cell on an Illumina NovaSeq sequencer.

Cell Ranger (v.3.0.1) (10X Genomics) was used for sample demultiplexing, barcode processing, unique molecular identifiers filtering, gene counting, and sample mapping to the reference transcriptome (mouse mm10 v.1.2.0).

**scRNA-seq analysis of mouse spinal cord microglia.** Raw sequencing data for each sample was converted to matrices of expression counts using the Cell Ranger software provided by 10X Genomics (version 3.0.2). Briefly, raw BCL files from the Illumina HiSeq were demultiplexed into paired-end, gzip-compressed FASTQ files using Cell Ranger's *mkfastq*. Using Cell Ranger's *count*, reads were aligned to the GRCm38 (mm10) mouse reference genome, and transcript counts quantified for each annotated gene within every cell. The resulting UMI count matrices (genes × cells) were then provided as input to Seurat suite (version 3.1.0). To filter out low-quality cells, we defined a window of a minimum of 500 and a maximum of 4000 detected genes per cell. Cells with more than 5% of the transcript counts derived from mitochondrial-encoded genes were further removed.

To identify sex-mediated differences in response to injury, all microglia samples were merged and analyzed using reciprocal PCA with reference-based integration as part of the Seurat single-cell analysis package[49,50]. Here we specify one or more of the datasets as the 'reference' for integrated analysis, with the remainder designated as 'query' datasets. In short, PCA was performed separately for each dataset following normalization, variable feature selection, and scaling. Integration anchors were identified using both male and female naïve samples as reference (*FindIntegrationAnchors* function with parameter *reduction = "rpca"*).

Clustering and visualization of the integrated dataset were performed using Uniform Manifold Approximation and Projection (UMAP), with the first 15 principal components at a resolution of 0.3. Differential expression analysis between two conditions was performed using *FindMarkers* with parameter *min.cells.group = 20* (minimum of 20 cells in any group being compared). DEGs were identified using the cutoffs: |log fold-change| > 0.25 and Benjamini-Hochberg FDR < 0.05. Cluster-specific marker genes were identified using similar cutoffs (*FindAllMarkers* function; upregulated genes only).

**Human spinal cord single-cell suspension.** Human intact spinal cord tissue was harvested from organ donors through a collaboration with Transplant Quebec. All procedures are approved by and performed in accordance with the ethical review board at McGill University (IRB#s A04-M53-08B). Familial consent for this study (IRB#s A04-M53-08B) was obtained for each subject. The rapid autopsy spinal cord samples were processed within 2 h of removal from the spinal column, and the tissue was kept on ice throughout. The samples were provided from the lumbar region of the cord (L1-L5) of three adults: two females and one male (see Supplementary Fig. 11). The tissue was processed as previously described[51]. Briefly, the meninges were carefully removed, and the tissue crosscut into small pieces of 1–2 mm$^3$. A single-cell suspension was obtained following mechanical and enzymatic (trypsin) dissociation. Myelin was removed using an isotonic Percoll gradient. Glia cells were located between the myelin and red blood cell layers, this layer was collected, washed, and resuspended in ice-cold PBS. Finally, a solution of cells (1000 cells per microliter) was sent for library preparation and high-throughput sequencing through Génome Québec. In accordance with the Single Cell 3′ Reagent Kits v2 User Guide (CG0052 10x Genomics), a single-cell RNA library was generated using the GemCode Single-Cell Instrument (10x Genomics, Pleasanton, CA, USA) and Single Cell 3′ Library & Gel Bead Kit v2 and Chip Kit (P/N 120236 P/N 120237 10x Genomics). The sequencing ready library was purified with SPRIselect, quality controlled for sized distribution and yield (LabChip GX Perkin Elmer), and quantified using qPCR (KAPA Biosystems Library Quantification Kit for Illumina platforms P/N KK4824). Finally, the sequencing was done using Illumina HiSeq4000 instrument. Cell barcodes and UMI (unique molecular identifiers) barcodes were demultiplexed and single end reads aligned to the reference genome, GRCh38, using the CellRanger pipeline (10X Genomics). The resulting cell-gene matrix contains UMI counts by gene and by cell barcode.

**Comparison of human spinal cord and human non-spinal (brain) microglia gene expression signatures.** To compare human spinal cord and human non-spinal (brain) microglia gene expression signatures, we used datasets from[19] (Masuda et al., accession code GSE:124335)[40], (Sankowski et al., accession code GSE:135437), and[41] (Olah et al., raw count matrix provided by the authors). Each dataset was analyzed separately with Seurat, including the SCTransform normalization method. For the Masuda et al., and Sankowski et al., studies, the average gene expression across all cells were considered for the downstream analysis. For Olah et al., study, the average expression of genes across homeostatic microglia (expressing CX3CR1 and C1QA genes) was considered. The expression values of the top 1000 highly expressed human spinal cord microglia genes were extracted from these studies and (log10 + 1) of the average expression values were used to draw the heatmap using ggplot2 package in R. The Venn diagram was obtained by comparing top 1000 highly expressed genes of each dataset.

**Cross-species analysis of single-cell transcriptomic data.** To compare transcription between species, we extracted one-to-one orthologs using BioMart software suite[52]. Human cells expressing either *TMEM119*, *ITGAM* or *CX3CR1* were included in the analysis. Mouse cells were subsampled to equal number of human cells, resulting in 5456 cells for each species. Mouse and human microglia samples were analyzed separately using Seurat (version 3.1.0). Briefly, canonical correlation analysis (CCA) was performed to identify shared sources of variation to produce anchors across the datasets, following *SCTransform* normalization. Clustering and visualization of the integrated dataset were performed using UMAP, using *FindClusters* and *RunUMAP* functions.

In order to compare global expression profiles of microglia across species, we aggregated all cell counts within each scRNA-seq sample to generate a pseudo bulk sample, and then analyze the pooled data by using approaches designed for bulk RNA-seq. For all downstream analyses, we excluded lowly expressed genes with an average read count lower than 10 across all samples, resulting in 10,672 genes in total. Raw counts were normalized using *edgeR*'s *TMM* algorithm[53] and were then transformed to log2-counts per million (logCPM) using the *voom* function implemented in the *limma* R package[54]. We tested for significantly DEGs using the *lmfit* function and nominal *p*-values were corrected for multiple testing using the Benjamini-Hochberg method.

Correlation heatmap was created by obtaining the Pearson correlation of logCPM expression values of all genes between samples. Pearson correlation coefficients were calculated using the *cor* function in R. UMI count data for neurons in the mouse dorsal horn were obtained from the GEO data repository (accession number GSE103840). Gene ontology analyses were conducted using Enrichr[55,56] for all datasets except Fig. 6, where gProfiler[57] was used. Pearson correlation coefficients between males and females were calculated using the log fold change (logFC) values of genes that are differentially expressed in either sex.

**Immunohistochemistry.** Mice were anesthetized and perfused intracardially with PBS followed by 4% paraformaldehyde (PFA). Spinal cords were removed from SNI/SHAM Day 3, Day 14, and 5 month post surgery mice, incubated in 4 % PFA at 4 °C for 24 h and then transferred to PBS. Spinal cords were cut transversely into 30 µm thick sections on a Vibratome. Sections were rinsed with PBS and blocked with 10% normal donkey/goat serum and primary antibodies were applied overnight (1:500 Ki67 [Abcam, ab15580], 1:500 Iba1 [Synaptic systems, 234 004], 1:500 Iba1 [Wako, 019-19741], 1:500 ApoE [Cell Signaling Technology, 13366]), 1:500 NeuN [Abcam, ab104224] and 1:500 GFAP [Abcam, ab4674]). Secondary antibodies (1:500 Goat anti-guinea pig Alexa Fluor 647 [Thermo Fisher Scientific, A-21450], 1:500 Donkey anti-rabbit Alexa Fluor 568 [Thermo Fisher Scientific. A10042], 1:500 Goat anti-guinea pig Alexa Fluor 488 [Thermo Fisher Scientific, A-11073], 1:500 Goat anti-mouse Alexa Fluor 405 [Thermo Fisher Scientific, A-31553], 1:500 Goat anti-chicken Alexa Fluor 647 [Thermo Fisher Scientific, A-32933]) were subsequently applied and tissue was mounted using ProLong Gold Antifade reagent [Thermofisher, P36934]. Images were acquired on a Zeiss LSM 880 confocal microscope equipped with an Argon multiline, 405 diode and HeNe 594 and 633 lasers. Each group consisted of a total of four mice, with 4–5 sections of spinal cord dorsal horn (ipsilateral) being taken from each mouse. Confocal stacks were taken at ×10 magnification (for Ki67), and ×63 oil immersion (for ApoE), and processed as maximum intensity projections of confocal z stacks using ImageJ software (v.1.53). ApoE images were analysed by calculating Mean Gray Value of respective channels using ImageJ. Ki67/Iba1 values were calculated by raw counting of Ki67 signal and Iba1 signal. TdTomato;TMEM119$^{CreERT2}$ pups (P4-P9) were intraperitoneally injected for five consecutive days with 20 µl of 50 mg/kg 4-hydroxytamoxifen (Sigma, H7904). Airyscan imaging was conducted using a Zeiss 63X/1.40 Oil DIC f/ELYRA objective and the Airyscan super-resolution (SR) module with 32-channel hexagonal array GaAsP detector on LSM880 (Zeiss).

**RNA in situ hybridization using RNAscope.** Mice were anesthetized and perfused intracardially with PBS follow by 4%PFA. Spinal cords were removed, incubated in 4% PFA at 4 °C for 1 h, incubated in 30% sucrose at 4 °C overnight and frozen at −80 °C for 24 h. Frozen spinal cords were cut transversely into 12 µm thick sections on a cryostat and stored at −80 °C until use. The RNAScope Multiplex Fluorescent V2 Assay (ACD Biosystems) was performed according to the ACD

protocol for fixed-frozen tissue. Spinal cord sections from naive male and female samples, as well as D3 and D14 SNI, were hybridized with one or two mRNA probes per experiment (*Gm3336* (1096051), *Spp1* (435191), *Ifit3* (508251), *Mki67* (416771), *Mcm6* (1096061), *Lgals1* (897151-C2), *Top2a* (491221), *Cldn5* (491611), *Pecam1* (316721-C2), *H2-EB1* (509081)). Microglia were genetically labeled by expressing tdTomato under the microglia-specific promoter, TMEM119 (tdTomato;TMEM119$^{CreERT2}$). The fluorophores Opal 520 and Opal 690 were used for labeling the probes. DAPI was used for nuclear counterstaining, and slides were mounted with ProLong Gold Antifade mounting medium (Life Technologies). Images were acquired on a Zeiss LSM 880 confocal microscope equipped with an Argon multiline, 405 diode and HeNe 594 and 633 lasers. Each group consisted of three mice, with two sections of spinal cord dorsal horn (ipsilateral) taken from each mouse. Confocal stacks were taken at ×20 magnification, and ×63 oil immersion, and processed as maximum intensity projections of confocal z stacks using ImageJ software (v.1.53).

**Human genetics**. Genetic analyses for the *APOE* gene were conducted in the UK Biobank[58,59]. Analyses were restricted to "White British" ancestry, after genotyping quality control performed by the UKB detailed in the resource: https://biobank.ctsu.ox.ac.uk/crystal/crystal/docs/genotyping_qc.pdf. Alleles for SNPs rs7412 and rs429358 were extracted using bgenix (bioArxiv 308296) and qctool (https://www.well.ox.ac.uk/~gav/qctool_v2/). The dosage data were converted into hard calls, using a threshold of 0.1 via PLINK[60]. Allelic combinations were assigned one of three possible haplotypes: ε2 (rs7412-T, rs429358-T), the ancestral haplotype ε3 (rs7412-C, rs429358-T), or ε4 (rs7412-C, rs429358-C). Association tests between haplotypes and pain phenotypes were performed using the haplo.cc function in the haplo.stats R computer package (https://cran.r-project.org/package=haplo.stats), with minimum counts of haplotypes set to 100, in an additive genetic inheritance model, and relative to the ancestral haplotype. Age, age squared, sex, recruitment sites, genotyping array, and first 40 principal genetic components were used as co-variables. Meta-analyses used an inverse standard-error based analytical strategy, as suggested by METAL[61]. Cases for pain at a body site were defined as participants answering "yes" to the following question: "In the last month have you experienced any of the following that interfered with your usual activities?" (UKB field 6159). Cases for chronic pain were defined as participants that answered also "yes" for the question: "Have you had [body site] pains for more than 3 months?". Body sites included: headaches (UKB field 3799, $n = 36{,}381$), facial (4067, $n = 3495$), neck/shoulder (3404, $n = 63{,}933$), stomach/abdominal (3741, $n = 18{,}741$), back (3571, $n = 70{,}634$), hip (3414, $n = 35{,}634$), knee (3773, $n = 68{,}237$) and widespread (2956, $n = 5{,}259$). Cases for acute pain were defined as participants that answered "no" to this question.

Body sites included: headaches ($n = 43{,}310$), facial ($n = 3882$), neck/shoulder ($n = 28{,}407$), stomach/abdominal ($n = 14{,}918$), back ($n = 32{,}209$), hip ($n = 9967$), knee ($n = 18{,}316$), and widespread ($n = 1038$). Control subjects were those that answered: "none of the above" at field 6159 ($n = 163{,}825$). Phenotype assignments for Alzheimer's diseases derived from UK Biobank field 20002 "Non-cancer illness code, self-reported" code 1263 "dementia/alzheimers/cognitive impairment", as well as main (field 41202) and secondary (field 41204) ICD10 diagnoses: G30 "Alzheimer's disease" and sub-codes (G300, G301, G308, G309) for a total of 470 cases and 384054 controls.

**Partitioning SNP heritability**. The stratified LD score regression formalism was used to estimate GWAS heritability from SNPs in genes expressed in specific microglia cell types[62,63]. To do so, we first performed a logistic regression between the sample's cluster membership (1 = yes, 0 = no) against its gene expression, with the sample's bio-specimen sex and the sample's average gene expression as co-variables. We specifically looked for positive but large values of the regression's test statistic, indicating that elevated gene expression was strongly associated with cluster membership, thus identifying genes most exclusively expressed by cells of the cluster. For each cluster, we retained the top 1500 genes with the largest test statistic values (about 10% of all genes, as suggested by Finucane and colleagues[63]). Genetic partitions per cluster were compiled using the complementary tool 'make_annot.py', with the top genes as input. Summary GWAS data were analyzed using 'ldsc.py' with the option '--h2-cts', enabling cluster-based partitioning of SNP heritability. The results consisted of heritability coefficient enrichment, its standard error, and associated *P*-value for enrichment in each cell type. Correction for multiple testing was accounted for by the false discovery rate (FDR). Correspondence between human gene symbols (HGNC[64]) and mouse ones (MGI[65]) was established using the BioMart[66] R package.

**Identification of pain-relevant, glia cluster-associated genes**. We identified genes simultaneously relevant to an acute/chronic pain site with a cluster-associated glia expression profile by (1) selecting genes with FDR at the 20% level in the GWAS, (2) selecting genes with FDR at the 20% level from the logistic regression's glia cell type expression specificity, and (3) selecting genes whose regression test statistic was positive. We did not consider the signs (positive or negative) of test statistics derived by MAGMA from summary GWAS data since they have no biologically interpretable meanings, but their magnitudes are indications of association strengths between the genes and the phenotypes. Information

about genes in the literature provided by the National Center for Biotechnology Information (NCBI's) GeneRIFs database[67].

**Statistical analysis**. All results are expressed as mean ± s.e.m. Statistical tests were made using a one-way ANOVA followed by between-group comparisons using Tukey's post hoc test or unpaired *t*-tests with $p < 0.05$ as the significance criterion (GraphPad Prism 9.3.0). Fisher's exact test was performed in R, using *fisher.test* function, to evaluate the statistical significance of the overlap between two gene lists.

**Reporting summary**. Further information on research design is available in the Nature Research Reporting Summary linked to this article.

## Data availability

Single-cell RNA-sequencing data generated in this study have been deposited in the Gene Expression Omnibus under the accession GSE162807. Publicly available datasets generated in previous scRNA-seq studies (Masuda et al.[19], Sankowski et al.[40], and Olah et al.[41]) were used in this work. Source data are provided with this paper.

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

## Acknowledgements

This study was supported by the Canadian Institutes of Health Research (PJT-162412) to A.K., and FRN-154281 to J.S.M. This work was also funded by the Canadian Excellence Research Chairs program to L.D. (CERC09). The current study was conducted under UK Biobank application no. 20802, application courteously initiated by Dr. Samar Khoury. S.T. was supported by a Natural Sciences and Engineering Research Council of Canada Ph.D. fellowship. A.U.G. was supported by the Mexican National Council for Science and Technology (CONACYT) and a Mitacs Globalink Graduate Fellowship Award. The flow cytometry work/cell sorting was performed in the McGill Flow Cytometry Core Facility; single-cell genomics at the McGill Genome Center, supported by Genome Canada and CFI (J.R.), and single-cell analysis in the Life Science Complex, supported by funding from the Canadian Foundation for Innovation. We thank Yu Chang Wang (McGill Genome Centre) for performing single-cell captures. We acknowledge Daniel Bisson for coordinating the rapid autopsy human spinal cord program.

## Author contributions

S. Tansley., S.U., A.U.G., L.H., J.S.M., and A.K. conceived the project, designed experiments, and supervised the research. S. Tansley., S.U., A.U.G., A. P., and J. R. performed mouse microglia analysis and M.Y., L.H., O.R., L.H., J.O., and C.S. performed human microglia analysis. M.P., and L.D. conducted human genetics studies. M.P.K., N.B., C.W., J.Z., S. Tahmasebi, and M.W.S. assisted with study design and interpretation of results. H.D., and A.R.S. assisted with IHC studies. S. Tansley., S.U., A.U.G., L.H., J.S.M., and A.K. wrote the manuscript. All authors reviewed the manuscript and discussed the work.

## Competing interests

The authors declare no competing interests.
