## [Peer Review File · Nature Communications]

Reviewer #1: (Remarks to the Author):

Microglia as the main innate immune cells in the CNS are known to modulate neuropathic pain in a gender-specific manner. However, microglia transcriptional signatures and states are not well defined during this condition. In this study, by using single-cell RNA sequencing the authors try to investigate the existence of microglia states that could promote or maintain neuropathic pain. While the approach and methodology are good, this reviewer has major concerns about data analysis, interpretation, and the conclusions made. In addition, several claims made by authors which are based on RNA-seq data and are not verified at all by complementary methods. There is lack of conclusive evidence to fully support authors claims and mechanisms of how ApoE could play a role chronic pain in missing. Thus, at this stage, the study is still immature.

Main points:

1) Figure 1: Authors state “Cells in all clusters expressed canonical microglia genes such as Tmem119, Fcrls, P2ry12, Cx3Cr1, Trem2, and C1qa”. The issue with this approach is that it will also include doublets, which consist of microglia and other cells.

We thank the reviewer for raising this important concern. In our study, we sorted microglia using FACS, gating for $CD45^{low}CD11B^{high}CX3CR1^{high}$, before performing scRNA-seq. During FACS, we isolated single cells by discriminating aggregates or doublets by gating the Forward Scatter-Width (FSC-W) low events in a Forward Scatter-Height (FSC-H) versus FSC-W bivariate plot. A second screening was then performed by selecting the Side Scatter-Width low (SSC-W) events in a Side Scatter-Height (SSC-H) versus SSC-W bivariate plot. We also sequenced far fewer cells than is theoretically possible in order to minimize doublet formation rates, and used the following criteria to filter out low-quality cells: (number of detected genes <500 or >4000 , number of UMI (RNA content) >15000 , and percent

mitochondrial reads >5). This approach was successfully used in previous scRNA-seq studies on microglia in the central nervous system. We also performed doublet analysis in our datasets using DoubletFinder (<https://github.com/chris-mcginnis-ucsf/DoubletFinder>)¹. DoubletFinder is a computational doublet detection method that integrates artificial doublets into existing scRNA-seq data and identifies real doublets as cells enriched for artificial nearest neighbors in gene expression space. As expected, this analysis showed that proliferating clusters (# 7 and 8) display increased incidence of doublets (24% and 29%, respectively, Rebuttal letter, Fig. 1a), whereas other clusters show smaller percentage of doublets (around ~7%), which is in the expected range based on 10X genomics guidelines (Rebuttal letter, Fig. 1b). Increased number of doublets in proliferating microglia clusters (cluster# 7 and 8) is likely a result of capturing cells at different stages of mitosis, as shown previously²⁻⁴. To directly study if doublets contribute to cluster markers in our analyses, we removed all doublets from our datasets using DoubletFinder, and then identified uniquely expressed markers for each cluster only for singlets. Correlation analysis between cluster markers before and after doublets removal for each cluster showed no change in cluster markers (correlation $r = 1$ for every cluster, see Rebuttal letter, Fig. 2). These analyses indicate that doublets most likely represent dividing cells or doublets formed between microglia (homotypic doublets) post-FACS isolation. Since a large proportion of proliferating microglia (cluster # 7 and 8), which we are interested in, would be affected by removing doublets and based on the lack of effect of doublets on cluster markers (Rebuttal letter, Fig. 2), we maintained our original datasets. However, in the revised version of the manuscript we have expanded the description of measures we implemented to reduce doubles using FACS before capturing cells for scRNA-seq analysis.

2) Several clusters show enrichment of non-microglial markers: Cluster 7 and 8 show enrichment of neuronal markers such as *Stmn1*, cluster 7 is enriched for expression of *Prc1* that is expressed by non-microglial cells (possible doublets). Cluster 10 shows expression of endothelial markers (doublets or contamination). It is not clear cluster 11 is microglia or CNS associated macrophages?

We apologize for not presenting and explaining our data more clearly. Fig. 1g shows unique markers that are enriched in each cluster as compared to other clusters. Therefore, they are uniquely defined markers within each microglia cluster as compared to all microglia. Not every cluster has unique markers. For example, there is no single marker gene that defines clusters 1, 2, and 3. Extended Data Fig. 5 shows genes with the highest expression level in each cluster. It does not mean that these genes are necessarily unique to any of the clusters or to microglia at all. Indeed, *Stmn1* is highly expressed in clusters 7 and 8

but is not unique to these clusters. Microglia in clusters 7 and 8 are proliferating microglia as cell cycle-associated genes, *Mki67* and *Top2a* are uniquely expressed in these two sub-populations and not in any other cluster (see Extended Data Fig. 6a) and show enrichment for cell division-related process in GO analysis (see Extended Data Fig. 6b). *Stmn1* regulates microtubule dynamics in many cellular processes, including cell division^{5,6} and cell motility. Because of its critical roles in the cell cycle, it is often referred to as the mitotic spindle regulator⁷. *Stmn1* has been shown to be highly expressed in microglia (see Fig. 1 in ⁸) and its levels are particularly upregulated in proliferating cells^{5,9}. As correctly indicated by the reviewer, *Stmn1* is also expressed in neurons¹⁰⁻¹², further demonstrating that this gene can be expressed in different cell types.

Similarly, PRC1 (Protein Regulator of cytokinesis) is involved in cell division (cytokinesis)¹³ in a variety of cell types. PRC1 is present at high levels during the S and G2/M phases of mitosis, but its levels drop dramatically when the cell exits mitosis and enters the G1 phase. Specifically, PRC1 is a member of the MAP65/ASE1 family of nonmotor microtubule-associated proteins. PRC1 is a substrate of CDK1, which maintains PRC1 in an inactive, monomeric state. Cell-cycle dependent degradation of CDK1 leads to dephosphorylation of PRC1 and subsequent KIF4-mediated translocation to the plus ends of microtubules, where it promotes microtubule bundling by cross-linking antiparallel microtubules. The microtubule bundling functions of PRC1 play a critical role in maintaining structural integrity of the spindle midzone during cytokinesis^{14,15}. Thus, the abundant expression of PRC1 in cluster 7 microglia is consistent with proliferating nature of cells in this cluster, and does not suggest that PRC1 is uniquely expressed in this subpopulation. In the revised version of the paper, we better explain the differences between uniquely expressed and abundantly expressed genes in each cluster.

We define cluster 10 as vessel-associated microglia since cells in this cluster express canonical microglia genes, along with unique expression of *Cldn5*. It has been previously shown that vessel-associated microglia express low levels of *Cldn5* at baseline and *Cldn5* expression is increased upon activation in this subpopulation¹⁶. As mentioned previously, removal of doublets did not affect cluster 10 markers (Rebuttal letter, Fig. 2), strengthening the notion that cells in this cluster uniquely express *Cldn5* and supporting categorization of this cluster as vessel-associated microglia. In the revised version of the manuscript, we performed *in situ* hybridization to show that *Cldn5* is present in microglia (labeled genetically in TMEM119^{CreERT2}tdTomato mice) in close proximity to endothelial cells (expressing *Pecam1* mRNA) (Extended data Fig. 7j), likely representing vessel-associated microglia.

Finally, we defined cluster 11 as perivascular macrophages as they express canonical microglia genes, although at reduced levels as compared to all other clusters (*Fcrls*, *Trem2*, *Cx3Cr1*, *Tmem119*, *Clqa* and *P2ry12*), along with unique expression of macrophage/monocyte markers (*H2-Aa*, *Mrc1*, *Ccr2*, *Lyve1*,

Dab2, *Mgl2*, *F13a1*) (Supplementary Table 1 and Extended Data Fig. 6d). Indeed, as shown in Extended Data Fig. 6d, macrophage-specific genes, *H2-Aa* and *Mrc1*, are expressed only in cluster 11 and not in any other cluster, strongly suggesting their macrophage identity. Interestingly, this subpopulation has been recently identified to play an important role in regulation of pain after peripheral nerve injury¹⁷. We have expanded the discussion on these cells in the revised version of the manuscript.

3) Does cluster 3 has a sub-cluster? Could this be a distinct cluster?

Indeed, UMAP distribution of cells in cluster 3 suggests that this cluster might be composed of several sub-clusters. However, additional formal sub-clustering analysis showed that cluster 3 can not be further divided into additional clusters based on the difference of 30% in the percentage of cells where the gene is detected in the cluster versus the rest of the cells.

4) How are top 8 genes selected per cluster? Is it unbiased or selected list? Showing a heat map of DEG will allow better visualization.

We apologize for not showing the heat map originally. Fig. 1h in the first submission showed top (most abundant) genes in each cluster and the selection of genes is unbiased. This list was taken from the accompanied table (Supplementary Table 2), showing genes with the highest expression in each cluster as compared to the entire dataset. As requested, we have now included a heatmap of the 8 most abundant genes per cluster (Extended Data Fig. 5) in the revised version of the manuscript.

5) Figure 2: The graphs a and b are not easy to visualize and the data point are not visible properly.

We have increased the size of the graphs in Fig. 2a and c, and made individual data points darker to improve visualization.

6) There is no data shown to claim that *Lgals1* and *Top2a* allow discriminating cluster 9 from other clusters.

We apologize for this oversight. Microglia in cluster 9 cannot be defined by a single marker gene. Moreover, a combination of two marker genes can only define a portion of cells in cluster 9. We have included the best 8 combinations of two cluster 9 markers that can be used to detect cells in this cluster

(Extended Data Fig. 7). Moreover, we performed *in situ* hybridization analysis for *Lgals1* and *Top2a* to localize cluster 9 microglia in the spinal cord (Extended Data Fig. 7h), showing that cluster 9 microglia are present in males but not in females post-SNI (Extended Data Fig. 7i).

7) Cluster 5 expresses several known DAM genes. However, this seems to be for naïve and SNI condition. Does this cluster represent microglia which are activated due to dissociation and sorting?

To address this comment and assess if DEGs in cluster 5 are enriched for extraction-related genes, we assessed whether known extraction-associated genes (Rebuttal letter, Fig. 3a) are enriched in cluster 5 or other clusters. Changes in the expression of extraction-associated genes in each cluster and in each condition (Rebuttal letter, Fig. 3b and c) showed that there is no enrichment for these genes in cluster 5, as compared to other clusters, in any condition in males and females.

8) Data for Cluster 7 and 8 are not convincing unless it is shown that these are not doublets.

As we showed in our response to point 1, removal of doublets did not affect markers for any cluster, indicating that doublets, which are formed most likely post-sorting, did not affect the definition of clusters. Moreover, microglia in clusters 7 and 8 uniquely express cell cycle-associated genes, *Mki67* and *Top2a* (Extended Data Fig. 6a) and show enrichment for cell division-related process in GO analysis (Extended Data Fig. 6b), altogether supporting the notion that these are proliferating microglia. Additional support for this conclusion comes from our observation that microglia in clusters 7 and 8 are present at very low level at the baseline, appear transiently at day 3 post-SNI, and are absent at later time points (day 14 and 5 months, Fig. 2a, c). This temporal profile corresponds to a) previous studies which showed that microglia proliferation peaks at day 3 post-injury and subsides thereafter¹⁸, and b) our own assessment of microglia proliferation dynamics at all time points (Fig. 2f-i). To support our immunohistochemistry results, we also performed *in situ* hybridisation for cluster 7/8 microglia using probes against *Mki67* (cluster 7) and *Mcm6* (cluster 8) (Extended Data Fig. 7d, e).

9) Cluster 9 is claimed to be male specific which appears at day 3. But authors do not include any data that complement RNA-seq data. Claims about the cluster 10 being vessel associated microglia are also not verified. Could authors cite studies which have shown this before and/or provide confirmation data as this cluster expresses other endothelial markers (Doublets?).

As requested by the reviewer, we performed many *in situ* hybridization experiments (Extended Data Fig. 7). Microglia were labelled using the expression of TdTomato under the microglia-specific promoter, TMEM119. We performed *in situ* hybridization on spinal cord sections from tdTomato;TMEM119^{CreERT2} mice for all clusters that can be defined by one unique marker gene (cluster 4, 5, 6, 7, 8, 10, and 11) and confirmed their colocalization within microglia in the dorsal horn spinal cord.

For cluster 10, we performed co-labeling of spinal cord sections from tdTomato;TMEM119^{CreERT2} mice with *Cldn5* (cluster 10 unique marker) and *Pecam1*, an endothelial marker, to show close association of cluster 10 microglia with blood vessels (Extended Data Fig. 7j). However, as correctly stated by the reviewer, our scRNA-seq analysis showed that this cluster is enriched for endothelial cell-specific markers. We therefore included a statement in the discussion section that we cannot rule out a cross-contamination of cluster 10 microglia with endothelial cells, as follows: “ We propose that cluster 10 microglia (0.19% of total microglia count) represent vessel-associated microglia. However, cells in this cluster express several endothelial gene (e.g., *Pecam1*). Thus, we cannot rule out the contamination of this subpopulation by endothelial cells during processing or incomplete dissociation from tightly associated endothelial cells. Alternatively, the presence of endothelial genes could be due to phagocytosis of endothelial cell fragments by microglia.”

For cluster 9, we used the best combination of two marker genes to detect cluster 9 microglia (*Lgals1⁺Top2a⁺*) and showed that these cells are found in males but not females at day 3 post-SNI (Extended Data Fig. 7i), consistent with our scRNA-seq analyses. We hope that these comprehensive analyses and more careful wording address the concerns of the reviewer.

10) Have authors verified increased microglia numbers at later times apart from day 3? Since authors do not show increased microglia upon proliferation, either numbers are increased at later stage or there are other mechanisms involved in balancing microglia numbers. Authors have not given any explanations.

As requested by the reviewer, we have performed quantification of all microglia (using Iba1) and proliferating microglia (using colocalization of Ki67 with Iba1) for all time points (naïve, day 3, day 14 and 5 months) and both sexes (Fig. 2h, i). Consistent with the previous report¹⁸, microglia proliferate shortly after the injury, peaking around day 3 post-SNI, and their proliferating rate is decreased at later time points. We also now provide several potential explanations for increased proliferation in males as compared to females without changes in total cell number, including a) different temporal dynamics of proliferation in males and females, b) increased microglia apoptosis in males, and c) enhanced recruitment of microglia from other areas of the spinal cord in females. Extensive studies, which are well

beyond the scope of this paper, would be required to test these options and identify the mechanism underlying this interesting phenomenon. We have initiated these studies, which might take another year or two (especially the differential migration of microglia in males and females) and hope to publish them in a separate follow-up manuscript.

11) Figure 3: Again authors do not show any primary data such as heat maps/dot plots to visualize DEGs across clusters. Showing the number of DEG does not provide meaningful information. It also seems that the changes that are described across different times are general across several microglia clusters and do not represent phenotype of one or few specific clusters of microglia that are associated with SNI. is this correct? In that case what is the importance of different microglia clusters? authors compare their microglia gene signatures with previously published signatures but it is difficult to understand the aim of this exercise in this paper.

Because of the large number of conditions (3 time points in males and females, 6 conditions total), and a large number of DEGs in each cluster (for example 336 DEGs in males and 219 in females just at day 3), it would be challenging to show primary data as heat maps/dot plots. However, all these data (DEGs per cluster per condition) are included in the accompanying supplementary tables (Supplementary Table 3). If the reviewer and editor feel these data would be better presented as several large heat maps/dot plots supplementary figures, we would be happy to do so.

As described in our paper, we found differential gene expression responses and altered cellular functions (based on GO analyses) in different clusters. Based on this information, we revealed that clusters 7 and 8 are proliferating cells, and this led us to discover that the proliferation is higher in males as compared to females. We confirmed this finding using immunohistochemistry against Ki67 in tdTomato;TMEM119^{CreERT2} mice. A cluster-based analysis also allowed us to identify robust and proinflammatory responses, strongly correlated to the previously described injury responsive microglia (IRM) signature, in cluster 9 microglia in males but not in females. Finally, we also identified a small population of perivascular macrophages, and their responses to nerve injury, which have been shown to play central roles in regulating microglia responses after nerve injury¹⁷. In summary, we believe that our work provides the long-awaited systematic characterization of microglial transcriptional states at the single cell level in response to peripheral nerve injury in mice at acute, sub-chronic and chronic phases, identifies the association between human *APOE* isoforms and distinct clusters of microglia with specific pain states in humans, reveals the enhanced microglial proliferation in male mice, and identifies a male-specific subpopulation of microglia. We think that our findings are important for the field of pain and

significantly advance the field toward deciphering the important and complex roles of microglia in neuropathic pain.

12) Figure 4: A clear rationale for choosing ApoE as a key gene is missing. It seems ApoE is not regulated in gender-specific manner. Also, this reviewer is confused about ApoE being the top DEG for cluster 5 (Figure 1) as in figure 4 it is shown that ApoE is the top DEG for several clusters. Could authors clarify this?

We apologize for being unclear regarding the rationale to focus on *ApoE*. *ApoE* is the most abundant gene in cluster 5, as shown in Fig. 1h. In addition, analysis of the DEGs after peripheral nerve injury (comparison between SNI and corresponding sham control) revealed that *ApoE* is the most upregulated DEG after SNI at two time points and in both sexes (Fig. 4). These results, together with known polymorphisms in the *APOE* gene and its link to numerous diseases in humans, including Alzheimer's disease, prompted us to study the association between *APOE* polymorphisms and pain in humans. This led to an exciting discovery that carriers of *APOE-ε4* are protected, whereas carriers of *APOE-ε2* have an increased risk to develop specific pain conditions. We have now also included a separate analysis for each sex (Fig. 5) which shows that the increased risk for chronic pain in carriers of *APOE-ε2* is detected only in men and not in women, indicating sex differences in humans. We are very excited about these findings and initiated a large study to investigate the role and the underlying mechanisms of each isoform (*APOE-ε2* and *APOE-ε4*) in pain. In the revised version of the manuscript, we better explain the rationale to focus on ApoE and the differences between ApoE being the most abundant gene in cluster 5 and the most upregulated DEG in several clusters after SNI.

13) Is there a difference between males and females in expression of ApoE? and if it is not the case then authors could simply state that ApoE is upregulated independent of gender instead of describing same thing two times.

We thank the reviewer for this suggestion. ApoE is upregulated in both male and female mice after SNI at chronic time points. We now modified the text to make it clearer.

14) ApoE immunostaining is not convincing as only one cell is shown per condition/time point. Authors should show overviews with better quality photomicrographs and provide quantifications of ApoE immunoreactivity.

As requested, we have performed numerous additional experiments and quantified ApoE immunoreactivity at all time points in males and females (Fig. 4f, g). We also used AiryScan microscopy to obtain high resolution images of ApoE and demonstrate its intracellular distribution pattern in microglia in chronic phases of neuropathic pain (Fig. 4e). We also present low- and high-magnification images in which ApoE is co-stained with markers of microglia, astrocytes, and neurons to show relative expression of ApoE in different cell types (Fig. 4d).

15) The human data with regards to co-relation between apoE alleles and chronic pain as well as human spinal cord microglia single cell analysis are interesting but clear connections between mouse and human data are missing and most of all mechanisms of how ApoE could play a role chronic pain are not addressed at all.

Indeed, our study identified an intriguing link between polymorphisms in the *APOE* gene and chronic pain in humans. Additionally, in the revised version of the manuscript, we performed a sex-based analysis and found that the association between *APOE* isoforms and pain in the human population is substantially stronger in men as compared to women (Fig. 5). We are very interested to understand the role of ApoE in pain and investigate the underlying mechanisms. We invested significant efforts to answer this important question within the timeframe of the review process. First, we generated AAV (AAV/TM6-EF1-mCherry-U6-LoxP-GFP-stop-LoxP-mApoE-shRNA, Vector Biolabs) with the goal to downregulate ApoE selectively in microglia (by expressing shRNA against ApoE in a Cre-dependent manner in microglia in TMEM119^{CreERT2} mice). Unfortunately, we could not show an abundant expression of *ApoE* shRNA, as assessed by the presence GFP, in microglia. For this AAV, we used an AAV serotype (AAV/TM6) that was claimed to have stronger tropism to microglia as compared to other serotypes¹⁹. Unfortunately, this approach was not efficient enough to allow strong expression of AAV in microglia and ablation of ApoE. In addition, we obtained *ApoE* general KO mice from The Jackson Laboratory. Analysis of behavioural phenotypes in two mouse models of nerve injury, SNI and CCI, showed no change in mechanical sensitivity at different time points. The lack of strong phenotypes in *ApoE* general KO mice might be caused by several factors including developmental compensation and redundancy with other lipoproteins. Importantly, this finding is consistent with studies in the Alzheimer's disease (AD) field, where ApoE is also upregulated. These studies showed that whereas humanized *APOE-ε4* mice have increased neuroanatomical (e.g., decreased hippocampus volume and dentate gyrus thickness, increased in astrocyte coverage in piriform cortex) deficits in a mouse model of AD, *ApoE* general KO

mice do not exhibit this phenotype²⁰.

[REDACTED]

We now include a statement that future studies on the functional role of ApoE in pain are required and clearly state that this is a limitation of the current work.

Minor points:

16) The references are incomplete: Line 64: Add Jordao M et al. Science 2019 for scRNAseq of myeloid cells in the CNS Line 141: Origin and fate of CNS macrophages were shown before by Goldmann T et al. Nat Immunol 2016 and by van Hove et al. Nat Neurosci 2019. Both references should be added.

We thank the reviewer for suggesting these important articles and as requested, added them to the revised version of the manuscript.

Reviewer #2 (Remarks to the Author):

In this paper authors undertake the first single cell RNAseq analysis of spinal microglia in mouse in the naïve state and following a nerve injury as well as comparison to human microglia. They also integrate

with human genetic findings in UK Biobank. There is a focus on sex specific effects which is a really critical issue in the pain field given the sex specific contribution of microglia to neuropathic pain previously shown by these authors (and others). They show microglia can be clustered into cell types and importantly there are male specific clusters which arise after injury (a male specific cluster of microglia at day 3) and also there are sex specific changes in differentially expressed genes and for instance at day 3 very different responses in the microglial clusters. This paper demonstrates novel and important findings relevant to microglia biology and sex differences in pain, these datasets will be an extremely valuable resource for glial and pain biologists (and these are being made available in GEO). Generally experiments are performed to a high standard. I particularly commend the inclusion of a very long time point (5 months) and integration with human scRNAseq. The supplementary data includes helpful methodological details (eg. FACS gating) and the Bioinformatics pipeline for scRNAseq looked appropriate to me. There are some issues regarding integration between human genetics and microglia.

Major issues:

1) Although I like the principle of using human genetic data as presented, I think it needs more thought:

a) The phenotypes used in UK-Biobank are acute and chronic pain phenotypes based on pain location with no information about aetiology of that pain. In as far as it goes I think this is reasonable, only a minority will be neuropathic. The ‘meta-analysis’ of conditions with a neuropathic component is misleading and should be removed. Neuropathic pain is defined by IASP as pain arising due to a lesion or disease of the somatosensory nervous system and certainly can’t be inferred from location. Even if you were to accept the premise that you can infer ‘neuropathic component’ from pain location (which I don’t) then decisions are arbitrary. All of these conditions said to NOT include a neuropathic component could have a neuropathic component: Facial pain- can be associated with trigeminal neuralgia (note these are self reported phenotypes for the most part participants won’t be able to distinguish causes of facial pain), headache uncommon but can be due to C2, 3 lesion, abdominal pain due to thoracic radiculopathy or proximal diabetic neuropathy, widespread pain secondary to small fiber neuropathy. In the conditions labelled as having a neuropathic component most of the causation will be non-neuropathic. The vast majority of shoulder, back, knee and hip pain will be osteoarthritis. To summarise the results regarding chronic pain and ApoE are interesting and should be retained but the neuropathic pain meta-analysis is misleading. There are other analyses that could be more useful in connecting the human and mouse data in terms of microglia (see below).

We agree with the reviewer and in the revised version of the manuscript, we removed the neuropathic meta-analysis, leaving the individual pain sites analysis (Fig. 5).

b) ApoE is not only expressed in microglia it is highly expressed by astrocytes and neurons. A more convincing argument regarding the role of microglia would be to identify microglia specific genes from the authors own (and other) scRNAseq datasets and integrate this data with GWAS signals arising from the chronic pain disorders to determine how variants in genes highly expressed in microglia are contributing to heritability. This type of workflow is now freely available in FUMA and would be a more convincing argument implicating microglia (as a cell type).

We thank the reviewer for this suggestion. Our study is focused on different microglial subtypes and their potential involvement in pain. To investigate whether genes in distinct mouse and human microglia clusters are enriched for pain-relevant genes identified in human GWAS, we substantially expanded our analyses and correlated highly expressed genes in each microglia cluster with pain GWAS UK biobank datasets. We revealed an association between transcriptional signature in several mouse and human microglia subtypes with distinct pain conditions (Extended Data Fig. 12). The strongest correlation included associations between the transcriptome of mouse microglia clusters 4 and 9 with GWAS genes associated with acute back pain (Extended Data Fig. 12b), and transcriptome of mouse cluster 2 microglia with genes linked to chronic stomach/abdominal pain. We also identified the top individual genes in several clusters contributing to the association (Extended Data Fig. 12n).

c) Rightly the authors repeatedly emphasise sex specific effects. Sex is a co-variate in the genetic analysis. Were there any sex specific effects in the gene variant associations?

We thank the reviewer for this suggestion, which prompted us to perform *APOE* genetic analyses separately for males and females, and indeed we discovered sex differences (Fig. 5). We found that the association between *APOE-ε4* and chronic pain (headache and back pain) is significant in both males and females. Surprisingly, whereas males show association between *APOE-ε2* and facial, abdominal, hip, and knee pain, no such association was detected in females. Moreover, males but not females show an association between *APOE-ε2* and acute headache. These data demonstrate sex-specific correlations between polymorphisms in the *APOE* gene and pain. We show these findings in Fig. 5 and discuss them in the revised version of the manuscript.

2) Regarding APoE expression. Figure 4- includes immunostaining of ApoE. Ideally immunostaining should be performed in female as well as male mice (I appreciate at transcript level expression is

increased in female but that does not mean it would be at protein level). We need lower power views and co-staining with other cell type markers- what is the pattern of ApoE expression in microglia relative to astrocytes, neurons etc. in the spinal cord. Finally what cellular compartment is it expressed in-? nuclear ?cytoplasmic (I didn't see convincing membrane/vesicular expression)?

To address this comment, we performed ApoE immunostaining in different conditions (SNI day 3, day 14 and 8 months in males and females) and quantified the expression levels of ApoE (Fig. 4f, g). The new data are consistent with our scRNA-seq results showing the upregulation of ApoE protein levels at day 14 and 8 months in both males and females.

We also performed co-immunostaining of ApoE with markers of microglia (Iba1), astrocytes (GFAP), and neurons (NeuN), and presented low magnification images to demonstrate relative expression of ApoE in different cell types (Fig. 4d).

Finally, using high-resolution AiryScan imaging, we show that ApoE is present in the cytoplasm of microglia as individual puncta (Fig. 4e), likely representing lipid particles²⁴.

Minor issues

1. Would it not have been possible to compare this scRNAseq dataset on spinal microglia in the naïve state with published datasets on brain microglia at least in broad terms to see if there are any key differences in sub-types? To my eye this is in itself an important question (I appreciate that there are already some comparisons with B Stevens published work following injury and batch effects may make it difficult to compare with published data but some discussion on whether the sub-groups are broadly similar would be helpful).

We thank the reviewer for this suggestion. We indeed have compared our data with previously identified transcriptional signatures (Fig. 3j, k). We have expanded the discussion on this topic as suggested by the reviewer.

2. Figure 6- correlations in panel F, are between mouse clusters 1-6 and human clusters 1-8. Human cluster 7 and 8 don't appear to have a 'correlate' in mouse. Although Human cluster 8 appears to relate to microglial proliferation does it bear any resemblance to mouse clusters 7, 8 which albeit small numerically but also seem to relate to proliferation and are not shown in this plot?

In the revised version of the manuscript, we expanded the correlation analysis to mouse clusters 1-8 and human clusters 1-8. This analysis showed that mouse cluster 8 exhibits correlation with several human microglia clusters, including cluster 8 (Fig. 6f).

3. Can the authors confirm that in harvesting spinal cord no attempt was made to separate ipsilateral versus contralateral or dorsal from ventral? Obviously the most relevant changes to explain pain will be ipsilateral dorsal to injury but there may have been methodological reasons why the whole lumbar segment was used in which case please just make this clear in methods.

To avoid dissection of the spinal cord to ipsilateral and contralateral sides, which could increase variability and introduce artifacts, mice underwent bilateral SNI/Sham and the whole lumbar spinal cord was collected for scRNA-seq analysis. The comparison was performed between SNI to control mice that underwent bilateral sham surgery. We have clarified this important experimental detail in the revised version.

Reviewer #3 (Remarks to the Author):

The manuscript by Tansley et al uses single cell RNA-seq to study the transcriptional responses in spinal cord microglia at three different time points after spared nerve injury in both male and female mice. They find several interesting differences between male and female microglial responses, including an increased proportion of proliferative microglia in male mice and a male/injury-specific microglia subset with an interferon/cytokine-induced signature. They also find that during the late time points after injury, ApoE was one of the most highly upregulated genes in both male and female microglia across several clusters. The authors go on to study the association of ApoE haplotype (e2, e3, e4) with chronic pain in a human cohort, finding an increased risk of long-term back and knee pain in apoe e2 carriers, and a decreased risk of back and knee pain and headaches in apoe e4 carriers. Finally the authors have performed single-cell RNA-seq of human microglia (n=3 cases) demonstrating the presence of several transcriptionally distinct clusters, including one highly expressing ApoE. The study is ambitious and well performed and there are several interesting findings that would be of interest to the field. The association between ApoE haplotype and chronic pain is particularly noteworthy. However, the study is entirely descriptive and does not address whether the sex and/or injury-specific microglia subsets contribute to different stages of pain development and maintenance (a question that is highly relevant to the field and the authors as it is asked in the second sentence of the abstract).

Major points:

1) The presence of a male and injury specific microglia subset (cluster 9) is probably the most interesting finding in the mouse data. Have the authors attempted to stain this subset in tissue using one or a combination of protein or mRNA-markers to address where it is distributed anatomically? Can this subset be found in other studies using single cell seq to study microglial responses to injury/inflammation (for example Masuda et al 2019, Hammond et al 2019)? In other words, could a comparative analysis be performed with the published literature to address whether it is a peripheral-nerve injury specific and/or spinal cord specific subset? It would also be interesting to understand whether the appearance of cluster 9 microglia is a transient response of homeostatic microglia or whether perhaps this subset differentiates into the ApoE+ microglia that are found at the later time points after nerve injury. It is interesting to note that cluster 5 and cluster 9 are very close in the UMAP. While it is not the expertise of this reviewer, it should be possible to bioinformatically analyze cell trajectories to better understand whether some clusters may in fact be different transition states of the same cells. See for example: <https://www.nature.com/articles/s41467-019-09670-4>. This type of analysis would be important not only as it relates to cluster 9 but for the entire dataset.

We thank the reviewer for their thoughtful suggestions. We performed *in situ* hybridization (RNAscope) experiments to localize clusters that can be defined by uniquely expressing markers in the spinal cord (Extended Data Fig. 7). Cluster 9 microglia can not be identified by one unique gene and a combination of inclusion and exclusion of gene markers ($Lgals1^{+}Top2\alpha^{-}$) is required, making the analysis challenging. Nevertheless, the RNAscope experiments confirmed our scRNA-seq results by showing that cluster 9 microglia are present in the dorsal horn at day 3 post-SNI in males but not in females. Importantly, we also show that gene expression pattern in cluster 9 microglia exhibits a very strong similarity to the previously defined transcriptional signature of injury-responsive microglia (IRM) (Fig. 3k). IRM were identified in the subcortical white matter in response to lysolecithin (LPC)-induced demyelination, a common model of multiple sclerosis (MS)²⁵. This finding suggests that cluster 9 microglia are not specific to peripheral nerve injury and can be also induced via direct damage to the central nervous system. As requested by the reviewer, we performed trajectory analysis for clusters 5 and 9 (Rebuttal letter, Fig. 4a-c) as well as for all clusters together (Rebuttal letter, Fig. 4d-f) for three post-injury time points. Despite significant efforts, unfortunately we were unable to identify clear patterns and come up with a coherent and clear picture to show transitions between subpopulations in different temporal phases. Since cluster 9 microglia are induced at day 3 post-SNI only in males, but ApoE is upregulated in numerous clusters at day 14 and 5 months in both sexes, we think it is unlikely that cluster 9 microglia

transition to ApoE-positive microglia. We can not, however, rule out the possibility that cluster 9 microglia transition to other less reactive state or alternatively, undergo apoptosis.

2) While the authors have certainly recorded important upregulations of ApoE across several microglia clusters at day 14 and 5 months post injury, they have not addressed whether this is a protective or detrimental response as it relates to chronic pain. Have the authors attempted to address the role of ApoE itself in the maintenance of chronic pain? Conditional deletion of ApoE in microglia would be the most relevant and interesting experiment, but this would be asking too much. Have the authors tested other approaches?

We agree with the reviewer that this is a very interesting and important scientific question that might have broad scientific and even clinical implications. As detailed in our response to the reviewer #1 (point# 15), we invested significant efforts and resources to study the functional role of ApoE in pain. Unfortunately, our studies using a viral approach and general *ApoE* KO mice were inconclusive.

[REDACTED]

3) It has been proposed that Trem2 and ApoE collaborate to drive the differentiation from a homeostatic microglial state into the DAM state (Krasemann et al 2017 Immunity, Keren-Shaul et al 2017 Cell). It would be interesting to understand whether the ApoE-responses observed in Figure 4 are dependent on TREM2.

We thank the reviewer for this suggestion. To address this comment using our datasets, we compared changes in the expression of *ApoE* and *Trem2* at different time points post-SNI (day 3, day 14 and 5 months) in each cluster in males and females. We found that changes in *ApoE* and *Trem2* are positively correlated in some clusters and conditions but negatively in others (Rebuttal letter, Fig. 5a for males and 6a for females). We also analysed *Trem2* and *ApoE* expression in different clusters (Rebuttal letter, Fig. 5b for males and 6b for females). This analysis showed that whereas *Trem2* is found in all microglia clusters, *ApoE* has a very distinct expression pattern. Thus, based on these analyses, we were unable to

unequivocally determine if the interaction between *Trem2* and *ApoE* drives microglia transition to disease states after nerve injury.

4. Cluster 10 microglia appear to be contaminated by endothelial cells or cell fragments. All top markers for this cluster found in Fig 1h (and additional ones in supp table 2, including *Pecam1* = pan-endothelial marker CD31) are highly expressed in brain endothelial cells. (see https://endotheliomics.shinyapps.io/ec_atlas/). It would be important to understand how the authors have excluded contamination from other CNS-cell types, which is a well-known problem when performing single-cell seq, particularly as it relates to the vasculature (see Vanlandewijck et al 2018 Nature <http://betsholtzlab.org/VascularSingleCells/database.html>). If the authors maintain that this is a true microglia subset it would be crucial to demonstrate expression of several of these markers histologically in microglia under naive conditions, as the given reference (Haruwaka et al) does not find *Cldn5*⁺ microglia under steady-state conditions.

To isolate microglia, we performed FACS, gating for *CD45*^{low}*CD11B*^{high}*CX3CR1*^{high}, before performing scRNA-seq. During FACS, we discriminated single cells from aggregates or doublets by gating the Forward Scatter-Width (FSC-W) low events in a Forward Scatter-Height (FSC-H) versus FSC-W bivariate plot. A second screening was then performed by selecting the Side Scatter-Width low (SSC-W) events in a Side Scatter-Height (SSC-H) versus SSC-W bivariate plot. This approach significantly reduces the chances of contamination by non-microglia cell types and has been used in similar studies. Additionally, clusters were defined based on the entire dataset (all cells in all conditions) and not only based on the transcriptome of microglia at baseline. Indeed, Haruwaka et al.¹⁶ indicate in their manuscript that the number of *Cldn5*⁺ microglia was low at baseline, but they still were able to identify a substantial population of *Cldn5*⁺ microglia. For example, in their Fig. 4d,c, the authors use IHC for *Cldn5* and *Iba1* to identify the expression of *Cldn5*⁺ in ~5% of parenchymal microglia and ~10% of vessel-associated microglia at baseline. The number of *Cldn5*⁺ vessel-associated microglia was increased post-LPS stimulation. To confirm this finding, they isolate microglia using FACS and show that in unstimulated WT mice, ~2.5% of all microglia express *Cldn5* (Supplementary figure 8b). In our dataset, cluster 10 microglia, which we think represent vessel-associated microglia, constitute a very small population of cells (0.19%). The difference between the abundance of *Cldn5*⁺ microglia in our study and the study of Haruwaka et al. might result from the assessment of different brain areas (spinal cord versus motor cortex) and/or differences in the sensitivity of detection methods (scRNA-seq versus IHC).

To address the concern of the reviewer experimentally, we performed *in situ* hybridization for *Cldn5* in mice in which microglia are genetically labeled (tdTomato;TMEM^{CreERT2}) and demonstrated the presence of *Cldn5*⁺, TMEM119⁺ microglia, consistent with our scRNA-seq data and the work of Haruwaka et al.¹⁶. However, we do agree with the reviewer that considering the presence of some other endothelial markers in cluster 10 microglia (that have not been described in vessel-associated microglia), we can not rule out cross-contamination by closely associated endothelial cells. We therefore indicate this limitation in the revised manuscript and mention that a potential contamination of cluster 10 microglia by endothelial cell or endothelial cell fragments (because of phagocytosis or experimental artifacts) can not be ruled out (lines 415-420).

5. Male microglia appear to respond more vigorously to nerve injury overall. By limiting their analysis to CD45^{low} microglia (as stated on row 103), could the authors have missed more heterogeneous microglia/myeloid responses in females? In other words, are there activated CD45^{high} microglia or monocytes/macrophages in female mice that were excluded from the analysis? The authors could perhaps use their FACS-data to see if the proportions of CD45^{high} cells change across the different experimental conditions. Along the same lines, female microglia proliferated less upon nerve injury, nonetheless the total number of dorsal horn microglia were not different between males and females (row 146-160). Could this be explained by infiltration of peripherally-derived macrophages? Analysis of TdTomato-Iba1⁺ vs TdTomato+Iba1⁺ cells in Tmem119CreER mice could help to address this question (Ext data Fig 4).

We followed the reviewer's suggestion and analysed the number of CD45^{high} cells in our FACS data. We did not find significant differences in the number of CD45^{high} cells between males and females in any condition (males versus females: Day 3, $p = 0.6429$, Day 14, $p = 0.9275$, 5 months, $p = 0.962$, unpaired t-test). Regarding the infiltration of peripheral macrophages, we performed our analysis using two different microglia labeling approaches: 1) Iba1, and 2) tdTomato;TMEM119^{CreERT2} mice. While Iba1 would also mark infiltrating macrophages, tdTomato;TMEM119^{CreERT2} is expressed selectively in spinal cord microglia and not in macrophages. The increased proliferation in males as compared to females was detected with both approaches, suggesting that the increased proliferation of microglia occurs in males, but the total number of microglia does not change. There are several potential explanations for this phenomenon. First, different dynamics of microglia proliferation, if faster in females, could lead to this outcome. An additional explanation is enhanced apoptosis of male microglia. Indeed, male microglia react to nerve injury more robustly, potentially leading to their cell death. A third potential explanation

is increased recruitment of microglia in females as compared to males. Female microglia show enhanced expression of motility genes. We now include all these potential scenarios in the discussion section of the revised manuscript.

6. It would be important that the authors more carefully compare their identified human microglial clusters to those already published, to better understand whether there may be spinal cord specific microglial subsets: (for example Sankowski et al 2019 Nature, Masuda et al 2019 Nature, Olah et al 2020 Nature Communications).

In the original submission, we compared clusters identified in our human spinal cord microglia dataset with clusters identified in the publications of *Hammond et al., 2019 Immunity* and *Keren-Shaul et al. 2017 Cell*. We have now followed the reviewer's suggestion to compare our spinal microglia signature to the published studies of non-spinal (brain) microglia datasets from Masuda et al. (accession code GSE:124335), Sankowski et al., (accession code GSE:135437), and Olah et al. (raw count matrix provided by the authors). Eight individual samples from the Masuda et al., study were processed separately and following removing low quality cells, merged into a single object containing 732 cells. This object was subjected to SCTransform normalization. Average gene expression of these cells was used for downstream analysis. Fifteen individual samples from the Sankowski et al., study were analyzed resulting in the average gene expression from 1,526 cells. For data from the Olah et al., study, the genobarcode count matrix of over 16,000 sequenced cells was provided by the authors. This matrix was subjected to the Seurat standard pipeline including SCTransform normalization and expression profiles of three homeostatic microglia clusters (7,801) cells were used for downstream analysis.

We first generated a heatmap (Extended Data Fig. 11c) considering the top 1000 highly expressed genes of our human spinal cord microglia and comparing the expression of these genes across all four datasets. In this preliminary analysis, we revealed that our spinal cord dataset is most similar to that of the microglia from Sankowski et al. with the Olah et al. microglia being the most distinct. The gene list with expression values of the analysed 1000 genes can be found in Supplementary Table 9. Next, we generated a Venn diagram (Extended Data Fig. 11d) using the top 1000 highly expressed genes of each dataset. Of these genes, 313 were common to all datasets with 250 (Spinal cord), 132 (Masuda) 136 (Sankowski) and 161 (Olah) genes unique to each dataset.

We believe these comparisons provide insights into the similarities and differences between human spinal cord and brain microglia, however, there are several important caveats to consider. First, microglia

analyzed across the studies have come from a variety of sources including autopsy and resected surgical tissue. Some of these microglia were sorted by flow cytometry prior to sequencing while others were sequenced as part of a mixed cellular suspension and the microglia were ‘isolated’ in silico. In addition to the varying numbers of cells analyzed and the depth of sequencing applied to each project, these cells were sequenced using different sequencing platforms (Extended Data Fig. 11e). The most appropriate experimental setup to answer this important question of a unique spinal cord microglia signature would involve the analysis of spinal cord and brain (non-spinal) tissue by the same group on the same sequencing platform. This is something we plan to do in the future.

Minor points:

1. It is difficult to assess how the cell proportions change across sex and injury from Fig 2a, b and c. Putting time on the x-axis and making combined or individual plots for each cluster is one suggestion. This could be combined with representative cluster color-coded UMAPs for each group, to see how cell densities change in clusters across the different conditions. If sham-groups are similar, it would be preferable to combine them.

We followed the reviewer’s suggestion and made changes to improve readability of the graphs in Fig. 2a-c. First, we made the graphs bigger and individual data points darker. Since our sham conditions are unique for each post-SNI time point and sex, we need to present them all. Additionally, we included in Extended Data Fig. 3 and 4, UMAP plots for each cluster and each condition. Since we have numerous conditions (7 for males and 7 for females for 11 clusters), including them in the main figures would be challenging.

2. The quantification of Ki67+ proliferative microglia in Figure 2f-g lacks a naive control. Ideally, day 14 and 5 month time points should also be included.

As requested, in the revised version of the manuscript, we quantify Ki67 in microglia in naïve control mice and at all three time points (day 3, day 14 and 5 months) (Fig. 2h).

3. It is unclear why Figure 4a, d and e are not presented separately for males and females.

We have now quantified ApoE levels separately in males and females and significantly expanded the IHC for ApoE, showing its expression at different time points (Fig. 4f, g).

4. Figure 4d. Please provide lower-power images so that apoe expression across several microglia and non-microglial cells can be visualized.

Harald Lund

Karolinska Institutet

We have significantly expanded the analysis of ApoE protein distribution. First, we show low magnification images of ApoE co-stained with a marker of microglia (Iba1), astrocytes (GFAP) and neurons (NeuN) (Fig. 4d). We also show ApoE immunostaining and quantification at all time points and both sexes (Fig. 4f, g). Finally, we present ApoE distribution in microglia using high resolution AiryScan imaging (Fig. 4e).

References:

- 1 McGinnis, C. S., Murrow, L. M. & Gartner, Z. J. DoubletFinder: Doublet Detection in Single-Cell RNA Sequencing Data Using Artificial Nearest Neighbors. *Cell Syst* **8**, 329-337 e324, doi:10.1016/j.cels.2019.03.003 (2019).
- 2 Mendes, M. S. *et al.* The role of P2Y12 in the kinetics of microglial self-renewal and maturation in the adult visual cortex in vivo. *Elife* **10**, doi:10.7554/eLife.61173 (2021).
- 3 Luecken, M. D. & Theis, F. J. Current best practices in single-cell RNA-seq analysis: a tutorial. *Mol Syst Biol* **15**, e8746, doi:10.15252/msb.20188746 (2019).
- 4 Sun, B. *et al.* Double-jeopardy: scRNA-seq doublet/multiplet detection using multi-omic profiling. *Cell Rep Methods* **1**, None, doi:10.1016/j.crmeth.2021.100008 (2021).
- 5 Rubin, C. I. & Atweh, G. F. The role of stathmin in the regulation of the cell cycle. *J Cell Biochem* **93**, 242-250, doi:10.1002/jcb.20187 (2004).
- 6 Cassimeris, L. The oncoprotein 18/stathmin family of microtubule destabilizers. *Curr Opin Cell Biol* **14**, 18-24, doi:10.1016/s0955-0674(01)00289-7 (2002).
- 7 Lush, M. E. *et al.* scRNA-Seq reveals distinct stem cell populations that drive hair cell regeneration after loss of Fgf and Notch signaling. *Elife* **8**, doi:10.7554/eLife.44431 (2019).
- 8 Yang, H. S. *et al.* Natural genetic variation determines microglia heterogeneity in wild-derived mouse models of Alzheimer's disease. *Cell Rep* **34**, 108739, doi:10.1016/j.celrep.2021.108739 (2021).

- 9 Ochocka, N. *et al.* Single-cell RNA sequencing reveals functional heterogeneity of glioma-associated brain macrophages. *Nat Commun* **12**, 1151, doi:10.1038/s41467-021-21407-w (2021).
- 10 Yamada, K. *et al.* Increased stathmin1 expression in the dentate gyrus of mice causes abnormal axonal arborizations. *PLoS One* **5**, e8596, doi:10.1371/journal.pone.0008596 (2010).
- 11 Carretero-Rodriguez, L. *et al.* The Rac-GAP alpha2-Chimaerin Signals via CRMP2 and Stathmins in the Development of the Ocular Motor System. *J Neurosci* **41**, 6652-6672, doi:10.1523/JNEUROSCI.0983-19.2021 (2021).
- 12 Fuller, H. R. *et al.* Stathmin is enriched in the developing corticospinal tract. *Mol Cell Neurosci* **69**, 12-21, doi:10.1016/j.mcn.2015.09.003 (2015).
- 13 Jiang, W. *et al.* PRC1: a human mitotic spindle-associated CDK substrate protein required for cytokinesis. *Mol Cell* **2**, 877-885, doi:10.1016/s1097-2765(00)80302-0 (1998).
- 14 Kurasawa, Y., Earnshaw, W. C., Mochizuki, Y., Dohmae, N. & Todokoro, K. Essential roles of KIF4 and its binding partner PRC1 in organized central spindle midzone formation. *EMBO J* **23**, 3237-3248, doi:10.1038/sj.emboj.7600347 (2004).
- 15 Mollinari, C. *et al.* PRC1 is a microtubule binding and bundling protein essential to maintain the mitotic spindle midzone. *J Cell Biol* **157**, 1175-1186, doi:10.1083/jcb.200111052 (2002).
- 16 Haruwaka, K. *et al.* Dual microglia effects on blood brain barrier permeability induced by systemic inflammation. *Nat Commun* **10**, 5816, doi:10.1038/s41467-019-13812-z (2019).
- 17 Niehaus, J. K., Taylor-Blake, B., Loo, L., Simon, J. M. & Zylka, M. J. Spinal macrophages resolve nociceptive hypersensitivity after peripheral injury. *Neuron* **109**, 1274-1282 e1276, doi:10.1016/j.neuron.2021.02.018 (2021).
- 18 Echeverry, S., Shi, X. Q. & Zhang, J. Characterization of cell proliferation in rat spinal cord following peripheral nerve injury and the relationship with neuropathic pain. *Pain* **135**, 37-47, doi:10.1016/j.pain.2007.05.002 (2008).
- 19 Rosario, A. M. *et al.* Microglia-specific targeting by novel capsid-modified AAV6 vectors. *Mol Ther Methods Clin Dev* **3**, 16026, doi:10.1038/mtm.2016.26 (2016).
- 20 Shi, Y. *et al.* ApoE4 markedly exacerbates tau-mediated neurodegeneration in a mouse model of tauopathy. *Nature* **549**, 523-527, doi:10.1038/nature24016 (2017).
- 21 Wagner, T., Bartelt, A., Schlein, C. & Heeren, J. Genetic Dissection of Tissue-Specific Apolipoprotein E Function for Hypercholesterolemia and Diet-Induced Obesity. *PLoS One* **10**, e0145102, doi:10.1371/journal.pone.0145102 (2015).

- 22 Huynh, T. V. *et al.* Lack of hepatic apoE does not influence early Abeta deposition: observations from a new APOE knock-in model. *Mol Neurodegener* **14**, 37, doi:10.1186/s13024-019-0337-1 (2019).
- 23 Wang, C. *et al.* Selective removal of astrocytic APOE4 strongly protects against tau-mediated neurodegeneration and decreases synaptic phagocytosis by microglia. *Neuron* **109**, 1657-1674 e1657, doi:10.1016/j.neuron.2021.03.024 (2021).
- 24 Wolfe, C. M., Fitz, N. F., Nam, K. N., Lefterov, I. & Koldamova, R. The Role of APOE and TREM2 in Alzheimer's Disease-Current Understanding and Perspectives. *Int J Mol Sci* **20**, doi:10.3390/ijms20010081 (2018).
- 25 Hammond, T. R. *et al.* Single-Cell RNA Sequencing of Microglia throughout the Mouse Lifespan and in the Injured Brain Reveals Complex Cell-State Changes. *Immunity* **50**, 253-271 e256, doi:10.1016/j.immuni.2018.11.004 (2019).

b

Multiplet Rate (%)	# of Cells Loaded	# of Cells Recovered
~0.4%	~825	~500
~0.8%	~1,650	~1,000
~1.6%	~3,300	~2,000
~2.4%	~4,950	~3,000
~3.2%	~6,600	~4,000
~4.0%	~8,250	~5,000
~4.8%	~9,900	~6,000
~5.6%	~11,550	~7,000
~6.4%	~13,200	~8,000
~7.2%	~14,850	~9,000
~8.0%	~16,500	~10,000

Rebuttal letter, Figure 1. Evaluation of doublets in microglia scRNA-seq data. (a) Percentage of doublets and singlets per cluster predicted using DoubletFinder (left), as well as a UMAP plot showing their distribution (middle). UMAP plot of mouse spinal cord microglia present in 11 distinct clusters (right). **(b)** Multiplet rate table of the 10x Genomics Single Cell 3' Gene Expression v3.1 assay.

Rebuttal letter, Figure 2. Correlation between cluster markers before and after doublets removal.

Correlation analysis between cluster markers before (singlets + doublets) and after doublets removal (only singlets) for each cluster. Pearson correlation coefficients (r) between cluster markers before and after removal of doublets were calculated using the log fold change (logFC) values of cluster markers.

a

DNAJB1
 DUSP1
 FOS
 FOSB
 H3F3B
 HSP90AB1
 HSP90B1
 HSPA1B
 HSPA5
 HSPA6
 HSPA8
 HSPB1
 IVNS1ABP
 JUN
 JUNB
 MALAT1
 RP11-212I21.4
 RPS16
 ZFP36L1
 HSPE1
 HSP90AA1
 HSPD1
 HSPH1
 SOCS3
 ZFP36
 EGR1
 JUND

b

Males

c

HIST1H4E
 HIST1H1C
 HIST1H2BG
 HIST1H2BN
 HIST1H2AC
 HIST2H2BE
 HIST2H2BF
 HIST1H1C
 HSPA1A

Females

Rebuttal letter, Figure 3. Expression changes of extraction-associated genes. (a) List of known extraction-associated genes. Changes in the expression of extraction-associated genes in each cluster, in each condition in males (b) and females (c).

Rebuttal letter, Figure 4. Trajectory analysis of mouse spinal cord microglia. (a) Trajectory analysis for clusters 5 and 9 showing the first two PCs. Trajectory analysis for clusters 5 and 9 showing the first two PCs color-coded by time points post-injury **(b)** and pseudotime **(c)**. **(d)** Trajectory analysis for the 11 distinct clusters showing the first two PCs. Trajectory analysis for the 11 distinct clusters showing the first two PCs color-coded by time points post-injury **(e)** and pseudotime **(f)**.

Rebuttal letter, Figure 5. Expression changes in *ApoE* and *Trem2* in male microglia. (a) Log fold change (logFC) values of *ApoE* and *Trem2* in each of the eleven clusters on day 3, day 14, and 5 months post-SNI. (b) UMAPS of *ApoE* in different conditions in male mice. Color codes for *ApoE* log-normalized counts. (c) UMAPS of *Trem2* in different conditions in male mice. Color codes for *Trem2* log-normalized counts.

a

b

c

Rebuttal letter, Figure 6. Expression changes in *Apoe* and *Trem2* in female microglia. (a) Log fold change (logFC) values of *Apoe* and *Trem2* in each of the eleven clusters on day 3, day 14, and 5 months post-SNI. (b) UMAPS of *Apoe* in different conditions in female mice. Color codes for *Apoe* log-normalized counts. (c) UMAPS of *Trem2* in different conditions in female mice. Color codes for *Trem2* log-normalized counts

Reviewers' Comments:

Reviewer #1: Remarks to the Author:

The authors largely addressed my questions and I have no additional comments.

Reviewer #2: Remarks to the Author:

I think that the authors have done a really good job in answering my queries in particular: the revised human genetic analysis showing sex specific effects and testing of association between transcriptional signature and human pain conditions. The new imaging of ApoE expression in microglia (and other cell types) is now convincing. I do not have any further concerns. I also agree with their comment that an in depth analysis of the contribution of ApoE to neuropathic pain (using humanised alleles etc) is interesting but outside the scope of this current manuscript.

Reviewer #3: Remarks to the Author:

The authors have done an excellent job responding to my questions. I only have one remaining concern, regarding major point 6 in my original remarks to the authors. Upon my request, the authors have performed a comparison of their human spinal cord microglia expression profile with that of three published studies (human brain microglia). However, it is my understanding that they have extracted average expression profiles for the datasets, essentially turning these single-cell studies into bulk-RNA data. I think it would be more interesting if the authors compared their identified microglia clusters with the microglia clusters found in these studies. I understand that a bioinformatic analysis would be challenging. Instead, the authors could read these studies in detail and compare and comment on overlap/similarity/novelty (comparing for example top genes, % of cells in cluster).